# *Ir56d*-dependent fatty acid responses in *Drosophila* uncover taste discrimination between different classes of fatty acids

**Elizabeth B Brown[1], Kreesha D Shah[1,2], Justin Palermo[1], Manali Dey[3], Anupama Dahanukar[3,4]\*, Alex C Keene[1]\***

[1]Department of Biological Sciences, Florida Atlantic University, Jupiter, United States; [2]Wilkes Honors College, Florida Atlantic University, Jupiter, United States; [3]Interdepartmental Neuroscience Program, University of California, Riverside, Riverside, United States; [4]Department of Molecular, Cell & Systems Biology, University of California, Riverside, Riverside, United States

**Abstract** Chemosensory systems are critical for evaluating the caloric value and potential toxicity of food. While animals can discriminate between thousands of odors, much less is known about the discriminative capabilities of taste systems. Fats and sugars represent calorically potent and attractive food sources that contribute to hedonic feeding. Despite the differences in nutritional value between fats and sugars, the ability of the taste system to discriminate between different rewarding tastants is thought to be limited. In *Drosophila*, taste neurons expressing the ionotropic receptor 56d (*IR56d*) are required for reflexive behavioral responses to the medium-chain fatty acid, hexanoic acid. Here, we tested whether flies can discriminate between different classes of fatty acids using an aversive memory assay. Our results indicate that flies are able to discriminate medium-chain fatty acids from both short- and long-chain fatty acids, but not from other medium-chain fatty acids. While *IR56d* neurons are broadly responsive to short-, medium-, and long-chain fatty acids, genetic deletion of *IR56d* selectively disrupts response to medium-chain fatty acids. Further, *IR56d+ GR64f+* neurons are necessary for proboscis extension response (PER) to medium-chain fatty acids, but both *IR56d* and *GR64f* neurons are dispensable for PER to short- and long-chain fatty acids, indicating the involvement of one or more other classes of neurons. Together, these findings reveal that *IR56d* is selectively required for medium-chain fatty acid taste, and discrimination of fatty acids occurs through differential receptor activation in shared populations of neurons. Our study uncovers a capacity for the taste system to encode tastant identity within a taste category.

**\*For correspondence:** anupama.dahanukar@ucr.edu (AD); keenea@fau.edu (ACK)

**Competing interests:** The authors declare that no competing interests exist.

## Introduction

Animals detect food primarily through taste and olfactory systems. Across phyla, there is enormous complexity in olfactory receptors and downstream processing mechanisms that allow for detection and differentiation between odorants (*Keller et al., 2017*; *Nara et al., 2011*; *Parnas et al., 2013*). By contrast, taste coding is thought to be simpler, with most animals possessing fewer taste receptors and a diminished ability to differentiate between tastants (*Freeman and Dahanukar, 2015*; *Scott, 2018*; *Yarmolinsky et al., 2009*). Most early studies in different species have focused on characterization of a limited number of taste modalities largely defined by human percepts (sweet, bitter, sour, umami, salt), though there is growing appreciation that additional taste pathways are likely to influence gustatory responses and feeding (*Chaudhari and Roper, 2010*; *Scott, 2018*). Between studies of *Drosophila* and mammals, cells or receptors that are involved in sensing water, carbonation, fat, electrophiles, polyamines, metal ions, and ribonucleotides have been identified,

suggesting a previously underappreciated complexity in the coding of tastants (*Cameron et al., 2010*; *Kang et al., 2010*; *Mishra et al., 2018*; *Zhang et al., 2013*). Elucidating the underlying mechanisms of tastant detection can provide fundamental insight into the molecular and cellular basis of tastant recognition and taste processing.

In flies and mammals, tastants are sensed by dedicated gustatory receptors that are expressed in gustatory receptor neurons (GRNs) or taste cells, respectively. In both systems, distinct subsets of taste sensory cells are responsive to compounds belonging to distinct taste modalities such as sweet or bitter, and convey information to discrete areas of higher order brain structures (*Vosshall and Stocker, 2007*; *Yarmolinsky et al., 2009*; *Zhang et al., 2003*). Given the conserved logic of taste processing, flies provide a powerful system for studying sensory processing and principles of taste circuit function (*Freeman and Dahanukar, 2015*; *Scott, 2018*; *Yarmolinsky et al., 2009*). Further, a number of genes and biochemical pathways that regulate feeding behavior are conserved across phyla (*Vosshall and Stocker, 2007*; *Yarmolinsky et al., 2009*). Notably, the gustatory system of *Drosophila* is amenable to in vivo Ca$^{2+}$ imaging and electrophysiology, both of which can be coupled with robust behavioral assays that measure reflexive taste responses and food consumption (*Wisotsky et al., 2011*). Taste neurons are housed in gustatory sensory structures called sensilla, which are located in the distal segments of the legs (tarsi), in the external and internal mouth organs (proboscis and pharynx), and in the wings. Each sensillum contains dendrites of multiple GRNs, each of which can be distinguished from the others based on its responses to various categories of tastants. Two main classes of non-overlapping gustatory neurons that have been identified are sweet-sensing and bitter-sensing neurons. Sweet-sensing GRNs promote feeding, whereas bitter-sensing GRNs act to deter (*Marella et al., 2006*; *Thorne et al., 2004*). Both sweet and bitter GRNs express subsets of 68 G-protein-coupled gustatory receptors (GRs) (*Clyne et al., 2000*; *Scott et al., 2001*). In addition, the *Drosophila* genome encodes 66 glutamate-like ionotropic receptors (IRs), a recently identified family of receptors implicated in taste, olfaction, and temperature sensation (*Benton et al., 2009*; *Rytz et al., 2013*). GRNs predominantly project to the subesophageal zone (SEZ), the primary taste center, but the higher order circuitry downstream of the SEZ contributing to taste processing is poorly understood (*Flood et al., 2013*; *Marella et al., 2012*; *Pool et al., 2014*; *Wang et al., 2004*). Determining how tastants activate GRNs that convey information to the SEZ and how these signals are transmitted to higher order brain centers is central to understanding the neural basis for taste and feeding.

In *Drosophila*, GRNs in the labellum and tarsi detect hexanoic acid (*Masek and Keene, 2013*). Mutation of ionotropic receptor 56d (*IR56d*) disrupts hexanoic acid taste, implicating *IR56d* as a fatty acid receptor, or as part of a complex involved in fatty acid taste (*Ahn et al., 2017*; *Sánchez-Alcañiz et al., 2018*). *IR56d* is co-expressed with *GR64f* (*Ahn et al., 2017*; *Tauber et al., 2017*), which broadly labels sweet GRNs (*Dahanukar et al., 2007*; *Jiao et al., 2008*; *Slone et al., 2007*). *IR56d*-expressing GRNs are responsive to both sugars and fatty acids, suggesting that these neurons may respond to diverse appetitive substances including multiple classes of fatty acids (*Tauber et al., 2017*). Notably, overlapping populations of sweet GRNs are responsive to different appetitive modalities and confer feeding behavior.

Are flies capable of differentiating between tastants of the same modality or is discrimination within a modality exclusively dependent on intensity? Taste discrimination can be assayed by training flies by pairing a negative stimulus with an appetitive tastant and determining whether the acquired aversion generalizes to another tastant (*Keene and Masek, 2012*; *Masek and Scott, 2010*). A previous study employing such experiments found that flies are unable to discriminate between different sugars (*Masek and Scott, 2010*). Conversely, we reported that flies can discriminate between sucrose (sugar) and hexanoic acid (fatty acid), revealing an ability to discriminate between appetitive stimuli of different modalities (*Tauber et al., 2017*). Here, we find that flies are capable of discriminating between different classes of fatty acids, despite broad tuning of fatty-acid-sensitive neurons to short-, medium-, and long-chain fatty acids.

## Results

Sugars and medium-chain fatty acids are sensed by an overlapping population of gustatory neurons, and flies can discriminate between these two attractive tastants (*Ahn et al., 2017*; *Tauber et al., 2017*). To test whether flies are capable of discriminating within a single modality, we measured the

ability of flies to discriminate between different types of fatty acids of the same concentration. We used an aversive taste memory assay in which an appetitive tastant applied to the proboscis is paired with application of bitter quinine immediately afterwards, resulting in an associative memory that inhibits responses to the appetitive tastant (*Masek et al., 2015*). A modified version of this assay, in which training with one tastant is followed by testing with another, allows us to determine whether flies can discriminate between these tastants (*Figure 1A*).

We first sought to determine whether flies are capable of differentiating between short- (3C–5C), medium- (6C–8C), and long-chain (>9C) fatty acids. We found that flies that were trained with pairing of quinine and medium-chain hexanoic acid (6C) exhibited proboscis extension response (PER) to subsequent application of short-chain valeric acid (5C; *Figure 1B*). Thus, aversive memory to 5C was not formed by training with 6C, suggesting that flies can discriminate between these short- and medium-chain fatty acids. Similarly, flies trained with 6C did not generalize aversive memory to 9C, consistent with the idea that flies can also discriminate between medium- and long-chain fatty acids (*Figure 1C*). To rule out the possibility that flies are unable to form aversive taste memories to short- and long-chain fatty acids, we trained with 5C and found robust aversive taste memory, which did not generalize to 9C (*Figure 1D*). Together, these results suggest that flies are capable of distinguishing between short-, medium-, and long-chain classes of fatty acids.

To determine whether flies can discriminate between compounds within each class of fatty acid, we tested the ability of flies to differentiate among each short-, medium-, and long-chain fatty acid class. We first trained flies to associate 4C, a short-chain fatty acid, with quinine, while a second short-chain fatty acid, 5C, was not reinforced. We found that the aversive memory formed with 4C was generalized to 5C, suggesting that flies cannot discriminate between different short-chain fatty acids (*Figure 1E*). Next, we trained flies to associate 6C, a medium-chain fatty acid, with quinine, while the medium-chain fatty acids 7C or 8C were not reinforced. In both cases, flies formed aversive memories to 6C, and these generalized to 7C and 8C, suggesting that flies cannot discriminate between different medium chain fatty acids (*Figure 1F, G*). To fortify these findings, we trained flies to 7C and measured the response to 8C. Again, flies formed aversive memory to 7C that was generalized to 8C (*Figure 1H*). Lastly, we trained flies to associate 10C, a long-chain fatty acid, with quinine, while a second long-chain fatty acid, 9C, was not reinforced. We again found that the aversive memory formed with 10C was generalized to 9C, suggesting that flies cannot discriminate among long-chain fatty acids (*Figure 1I*).

To verify that taste discrimination observed between different classes of fatty acids is not simply the result of prior experience, we performed reciprocal experiments to those in *Figure 1*, in which the tested tastant was used during training, and the trained tastant was used to assess taste discrimination. Consistent with the findings from the first series of experiments, we found that flies are able to discriminate between short-, medium-, and long-chain fatty acids (*Figure 1—figure supplement 1A–C*), but are unable to discriminate between fatty acids within a particular class (e.g., medium-chain fatty acids; *Figure 1—figure supplement 1D–F*). Further, it is possible that repeated presentation of fatty acid may alter the valence of the tastant over time. To address this, we asked whether there were any differences in PER between the pretest, training, and test applications among the naïve groups in each test of taste discrimination. We found that PER to consecutive applications of fatty acid shows no significant change in responsiveness over time, although in some cases PER to fatty acid trends downward (*Figure 1—figure supplement 1*). Taken together, these results reveal that flies cannot discriminate among different short-, medium-, or long-chain fatty acids, although they are able to discriminate medium-chain fatty acids from short- or long-chain fatty acids.

A potential confounding factor in the taste memory assay used to assess discrimination is that flies may discriminate based on perceived intensity rather than the class identity of fatty acids. To test this possibility, we tested whether flies trained to 1% 6C could discriminate a different fatty acid tested at a different concentration (0.1%). Even under these test conditions, flies remained able to discriminate between short- (5C) and medium-chain (6C) fatty acids as well as between long- (9C) and medium-chain (6C) fatty acids (*Figure 1—figure supplement 2A, B*). However, conditioning to a medium-chain fatty acid (6C) generalized to another medium-chain fatty acid (8C), suggesting that despite a 10× difference in concentration, flies remain unable to distinguish between medium-chain fatty acids (*Figure 1—figure supplement 2C*). These results fortify the notion that flies are able to distinguish between short-, medium-, and long chain fatty acids, but not within fatty acids of the same class.

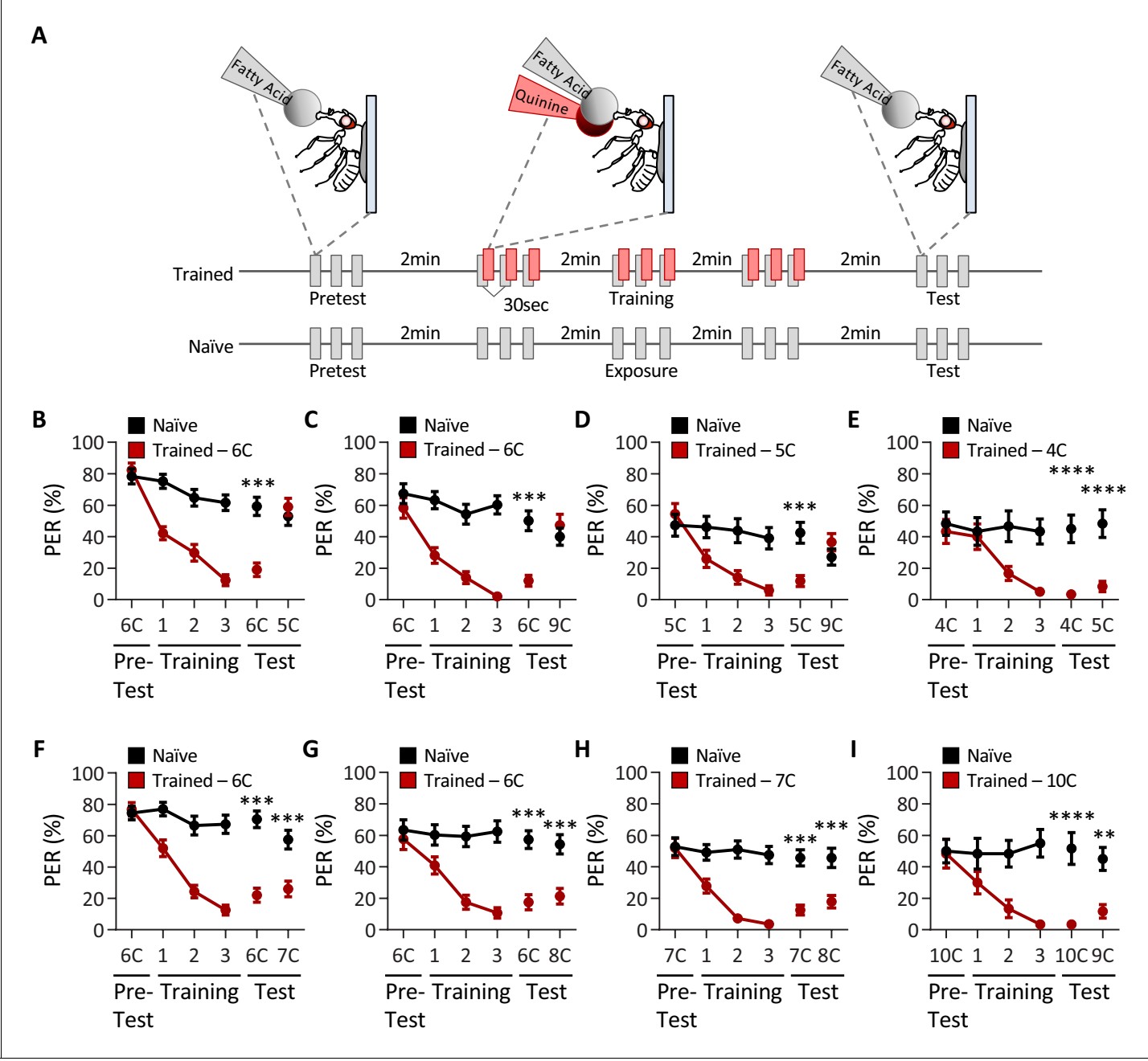

**Figure 1.** *Drosophila* can discriminate between, but not among, short-, medium-, and long-chain fatty acids. (**A**) An aversive taste memory assay was used to assess fatty acid taste discrimination in female $w^{1118}$ flies at a 1% concentration. First, initial responses to a short-, medium-, or long-chain fatty acid were assessed (Pretest). Next, flies were trained by pairing this fatty acid with quinine presentation immediately following tastant application (Training). Proboscis extension response (PER) in response to either the same or different fatty acid was then tested in the absence of quinine (Test). In control experiments (Naïve), the same procedure was followed, but quinine was not applied to the proboscis. (**B**) The pairing of medium-chain hexanoic acid (6C) and quinine (red) results in a significant reduction in PER compared to naïve flies. After training, PER response to 6C was significantly lower in trained flies compared to naïve flies (p<0.0001), but there was no difference in PER to short-chain valeric acid (5C; p=0.6864). Restricted maximum likelihood (REML): $F_{1,80} = 7.329$, p=0.0003, with Sidak's test for multiple comparisons; N = 40–42. (**C**) The pairing of medium-chain hexanoic acid (6C) and quinine (red) results in a significant reduction in PER compared to naïve flies. After training, PER response to 6C was significantly lower in trained flies compared to naïve flies (p<0.0001), but there was no difference in PER to long-chain nonanoic acid (9C; p=0.3346). REML: $F_{1,64} = 6.296$, p=0.0146, with Sidak's test for multiple comparisons; N = 33. (**D**) The pairing of short-chain valeric acid (5C) and quinine (red) results in a significant reduction in PER compared to naïve flies. After training, PER response to 5C was significantly lower in trained flies compared to naïve flies (p=0.0014), but there was no difference in PER to long-chain nonanoic acid (9C; p=0.0789). REML: $F_{1,46} = 2.721$, p=0.0105, with Sidak's test for multiple comparisons; N = 24. (**E**) The pairing of short-chain butanoic acid (4C) and quinine (red) results in a significant reduction in PER compared to naïve flies. After training, PER to

*Figure 1 continued on next page*

Figure 1 continued

both 4C and short-chain valeric acid (5C) was significantly lower in trained flies compared to naïve flies (4C: p<0.0001; 5C: p<0.0001). REML: $F_{1,38}$ = 33.67, p<0.0001, with Sidak's test for multiple comparisons; N = 20. (F) The pairing of medium-chain hexanoic acid (6C) and quinine (red) results in a significant reduction in PER compared to naïve flies. After training, PER to both 6C and medium-chain heptanoic acid (7C) was significantly lower in trained flies compared to naïve flies (6C: p<0.0001; 7C: p<0.0001). REML: $F_{1,81}$ = 45.88, p<0.0001, with Sidak's test for multiple comparisons; N = 41–42. (G) The pairing of medium-chain hexanoic acid (6C) and quinine (red) results in a significant reduction in PER compared to naïve flies. After training, PER to both 6C and medium-chain octanoic acid (8C) was significantly lower in trained flies compared to naïve flies (6C: p<0.0001; 8C: p<0.0001). REML: $F_{1,65}$ = 32.76, p<0.0001, with Sidak's test for multiple comparisons; N = 33–34. (H) The pairing of medium-chain heptanoic acid (7C) and quinine (red) results in a significant reduction in PER compared to naïve flies. After training, PER to both 7C and medium-chain octanoic acid (8C) was significantly lower in trained flies compared to naïve flies (7C: p<0.0001; 8C: p<0.0001). REML: $F_{1,72}$ = 33.67, p<0.0001, with Sidak's test for multiple comparisons; N = 37. (I) The pairing of long-chain decanoic acid (10C) and quinine (red) results in a significant reduction in PER compared to naïve flies. After training, PER to both 10C and long-chain nonanoic acid (9C) was significantly lower in trained flies compared to naïve flies (10C: p<0.0001; 9C: p=0.0015). REML: $F_{1,38}$ = 33.23, p<0.0001, with Sidak's test for multiple comparisons; N = 20. Error bars indicate ± SEM. **p<0.01; ***p<0.001; ****p<0.0001.

The online version of this article includes the following source data and figure supplement(s) for figure 1:

**Source data 1.** Raw taste discrimination data between short-, medium-, and long-chain fatty acids.

**Figure supplement 1.** *Drosophila* can discriminate between, but not among, short-, medium-, and long-chain fatty acids.

**Figure supplement 2.** Tastant intensity does not affect the ability of *Drosophila* to discriminate between short-, medium-, and long-chain fatty acids.

**Figure supplement 3.** *Drosophila* of both sexes are responsive to short-, medium-, and long-chain fatty acids.

**Figure supplement 4.** *Drosophila* males can discriminate between short-, medium-, and long-chain fatty acids, but not among medium-chain fatty acids.

We next sought to determine whether the discrimination observed in female flies is also found in males. PER analysis with a panel of short-, medium-, and long-chain fatty acids revealed that males respond to all fatty acids tested, though the overall responses were generally lower than those observed in females (*Figure 1—figure supplement 3*). To assess whether male flies are able to discriminate between different classes of fatty acids, we trained flies to a medium-chain fatty acid (6C) and then measured discrimination between 4C (short), 8C (medium), and 9C (long) fatty acids. Similar to results obtained in female flies, males were able to discriminate between 6C and 4C as well as between 6C and 9C, but not between 6C and 8C (*Figure 1—figure supplement 4*). Therefore, male flies are also able to discriminate different classes of fatty acids, but are not able to distinguish fatty acids within the same class.

The short-, medium-, and long-chain fatty acids that we tested have distinctly different olfactory activation profiles (*Hallem and Carlson, 2006*), raising the possibility that flies can discriminate between these compounds using a combination of olfactory and gustatory information. To exclude the effects of olfactory input, we surgically ablated the antennae, the maxillary palps, or both structures, and measured the ability of flies to discriminate between a representative tastant from each short-, medium-, or long-chain fatty acid (*Figure 2A*). All test groups were able to distinguish between sucrose and hexanoic acid, confirming that the ablation itself does not generally impact taste or memory formation (*Figure 2B*). Further, flies trained to a medium-chain fatty acid (6C) did not generalize aversion to short- (5C) or long-chain (9C) fatty acids, regardless of the absence of one or both olfactory organs (*Figure 2C, D*). Taken together, our findings reveal an ability of the taste system to encode the identity of different classes of fatty acids.

In previous work, we reported that *IR56d* neurons are required for 6C taste (*Tauber et al., 2017*). The finding that flies cannot discriminate between medium-chain fatty acids raises the possibility that *IR56d* neurons mediate taste of medium-chain fatty acids, but not short- and long-chain fatty acids. To test this possibility, we silenced *IR56d*-expressing neurons using the synaptobrevin cleavage peptide tetanus toxin light chain (TNT; *Sweeney et al., 1995*) and measured PER to multiple classes of fatty acids, including short-, medium-, and long-chain fatty acids (*Figure 3A*). To control for any non-specific effects of TNT, we compared PER in flies with silenced *IR56d*-expressing neurons (*IR56d*-GAL4 > UAS TNT) to flies expressing the inactive variant of TNT in *IR56d*-expressing neurons (*IR56d*-GAL4 > UAS impTNT). Consistent with previous findings (*Tauber et al., 2017*), we observed no effect of silencing *IR56d*-expressing neurons on PER to sucrose (*Figure 3B*). Next, we measured PER to a panel of saturated fatty acids ranging from 4C (butanoic acid) to 10C (decanoic acid) in length (*Figure 3C*). Control flies exhibited a robust PER to all seven fatty acids, revealing that at least at a 1% concentration many diverse classes of fatty acids can trigger this behavioral response.

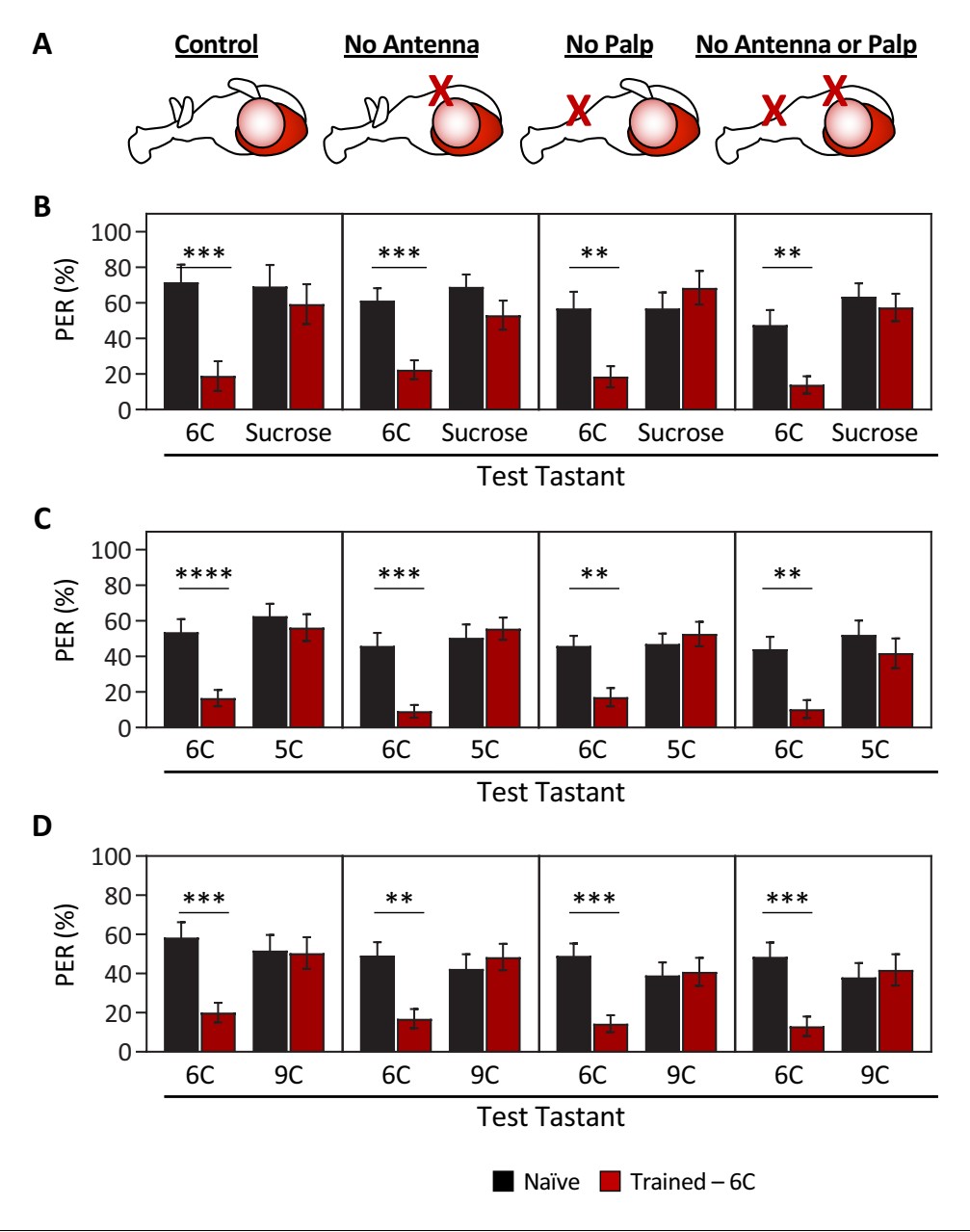

**Figure 2.** Ablation of olfactory organs has no effect on the ability of *Drosophila* to discriminate between short-, medium-, and long-chain fatty acids. Aversive taste memory was measured as described in *Figure 1A*. Flies were trained by pairing 1% medium-chain hexanoic acid (6C) with quinine (Training; see *Figure 1—figure supplement 3*) and then proboscis extension response (PER) in response to either 10 mM sucrose, 1% short-chain valeric acid (5C), or 1% long-chain nonanoic acid (9C) was measured in the absence of quinine (Test). (A) Aversive taste memory was measured in unmanipulated control flies (first panel), in flies without antennae (second panel), maxillary palps (third panel), or both antennae and maxillary palps (fourth panel). (B) For all ablation treatments, taste memory to medium-chain hexanoic acid (6C) was significantly lower in trained flies compared to naïve flies, but there was no difference in PER to sucrose. Restricted maximum likelihood (REML): $F_{1,86}$ = 42.41, p<0.0001, with Sidak's test for multiple comparisons; N = 13–26. (C) For all ablation treatments, taste memory to 6C was significantly lower in trained flies compared to naïve flies, but there was no difference in PER to short-chain valeric acid (5C). REML: $F_{1,103}$ = 51.87, p<0.0001, with Sidak's test for multiple comparisons; N = 19–31. (D) For all ablation treatments, taste memory to 6C was significantly lower in trained flies compared to naïve flies, but there was no difference in PER to long-chain nonanoic acid (9C). REML: $F_{1,97}$ = 11.47, p=0.0010, with Sidak's test for multiple comparisons; N = 22–27. Error bars indicate ± SEM. **p<0.01; ***p<0.001; ****p<0.0001.
*Figure 2 continued on next page*

*Figure 2 continued*
The online version of this article includes the following source data for figure 2:
**Source data 1.** Raw taste discrimination data after ablation of olfactory organs.

To determine whether *IR56d* neurons are generally required for detection of fatty acids or selectively required for sensing hexanoic acid, we next measured PER in flies with *IR56d*-expressing neurons silenced. Silencing *IR56d*-expressing neurons significantly reduced PER to the three medium-chain fatty acids (6C, 7C, and 8C). Conversely, there was no difference in PER between control and *IR56d*-silenced flies in response to short-chain (4C and 5C) and long-chain (9C and 10C) fatty acids.

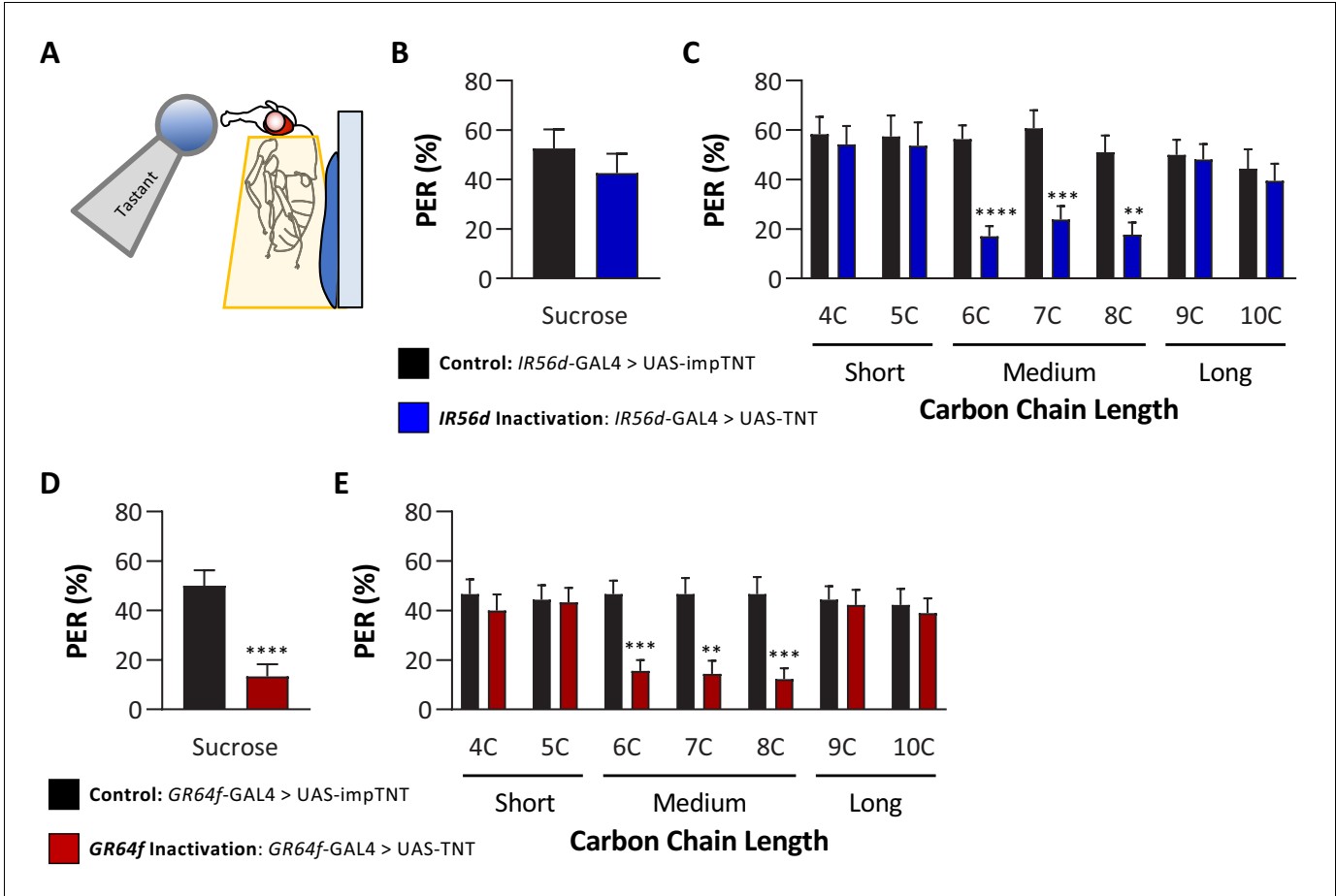

**Figure 3.** Silencing both *IR56D*- and *GR64f*-expressing neurons reduces taste perception to medium-chain fatty acids. (**A**) Proboscis extension response (PER) was measured in female flies after 24 hr of starvation. Either 10 mM sucrose or 1% fatty acid was applied to the fly's labellum for a maximum of 2 s and then removed to observe proboscis extension reflex. (**B**) Blocking synaptic release by genetic expression of light-chain tetanus toxin (UAS-TNT) in *IR56D*-expressing neurons has no effect on PER to sucrose compared to control flies expressing an inactive form of tetanus toxin (UAS-impTNT). Mann–Whitney test: U = 595, p=0.8410; N = 35. (**C**) Silencing *IR56D*-expressing neurons significantly reduces PER to medium-chain fatty acids (6C–8C), but has no effect on PER to either short- (4C, 5C) or long-chain fatty acids (9C, 10C). Restricted maximum likelihood (REML): $F_{1,406}$ = 25.03, p<0.0001, with Sidak's test for multiple comparisons; N = 24–45. (**D**) Blocking synaptic release by genetic expression of light-chain tetanus toxin (UAS-TNT) in *GR64f*-expressing neurons significantly reduces PER to sucrose compared to control flies expressing an inactive form of tetanus toxin (UAS-impTNT). Mann–Whitney test: U = 177, p<0.0001; N = 30. (**E**) Silencing *GR64f*-expressing neurons significantly reduces PER to medium-chain fatty acids (6C–8C), but has no effect on PER to either short- (4C, 5C) or long-chain fatty acids (9C, 10C). REML: $F_{1,58}$ = 22.68, p<0.0001, with Sidak's test for multiple comparisons; N = 30. Error bars indicate ± SEM. **p<0.01; ***p<0.001; ****p<0.0001.
The online version of this article includes the following source data and figure supplement(s) for figure 3:

**Source data 1.** Raw data from proboscis extension response experiments to short-, medium-, and long-chain fatty acids.
**Figure supplement 1.** *IR76b* and *IR25a* are required for taste perception to short-, medium-, and long-chain fatty acids.

Therefore, *IR56d*-expressing neurons are required for medium-chain fatty acid taste perception, but are dispensable for PER to both short- and long-chain fatty acids.

Since *IR56d* is required for taste response to medium-chain, but not short- or long-chain fatty acids, it is possible that other IRs mediate this response. As a first step in identifying which additional receptor(s) may be involved, we measured PER in both *IR76b* and *IR25a* mutants to short-, medium-, and long-chain fatty acids as these broadly expressed receptors have been previously found to mediate taste response to medium-chain fatty acids (*Ahn et al., 2017*). In agreement with these findings, PERs to medium-chain fatty acids were significantly reduced for both the *IR76b* and *IR25a* mutants, while PER to sucrose was normal (*Figure 3—figure supplement 1*). Additionally, we found that PERs to both short- and long-chain fatty acids were also significantly reduced in both mutants, suggesting that both *IR76b* and *IR25a* are required for taste response to all three classes tested (*Figure 3—figure supplement 1*).

The finding that *IR56d*-expressing neurons are required selectively for PER to medium-chain fatty acids raises the possibility that other subsets of sweet-sensing *GR64f*-expressing neurons are required for PER to short- (4C, 5C) and long-chain (9C–10C) fatty acids. Alternatively, it is possible that these classes of fatty acids elicit PER through gustatory neurons that are not labeled by *GR64f*. To differentiate between these possibilities, we silenced *GR64f*-expressing neurons and measured PER to short-, medium-, and long-chain fatty acids. In agreement with previous findings, silencing of *GR64f*-expressing neurons significantly reduced PER to sucrose (10 mM) as well as to medium-chain fatty acids (*Figure 3D, E*). However, we found no significant differences in responses to both short- and long-chain fatty acids between *GR64f*-silenced (*GR64f*-GAL4 > UAS TNT) and control (*GR64f*-GAL4 > UAS impTNT) flies. Therefore, PER elicited by short- and long-chain fatty acids is conferred by neurons that express neither *IR56d* nor *GR64f*.

To directly assess whether the *IR56d* receptor mediates responses to medium-chain fatty acids, we used the CRISPR/Cas9 system to generate an *IR56d* allele in which a GAL4 element is inserted into the *IR56d* locus (*IR56d*[GAL4]; *Figure 4A*), thereby allowing expression of UAS-transgenes under the control of the *IR56d* promoter. To confirm that the GAL4 knock-in element is indeed expressed in *IR56d*-expressing neurons, we generated flies carrying both UAS-mCD8:GFP and the *IR56d*[GAL4] allele (*IR56d*[GAL4]>UAS-mCD8:GFP) and mapped the expression of GFP. Consistent with previous findings, we found GFP expression in labellar neurons that projected axons to both the taste peg and sweet taste regions of the SEZ (*Figure 4B–E*; *Koh et al., 2014*; *Tauber et al., 2017*). In agreement with previous findings from genetic silencing of *IR56d*-expressing neurons, PER to sucrose did not differ between *IR56d*[GAL4] and control flies (*Figure 4F*), suggesting that *IR56d*[GAL4] is dispensable for response to sucrose. To examine the role of *IR56d* in fatty acid taste, we measured PER to fatty acids ranging from 4C to 10C in length (*Figure 4G*). We found that PER to medium-chain fatty acids was disrupted in *IR56d*[GAL4] flies (6C–8C), whereas PER to short- (4C and 5C) and long-chain fatty acids (9C and 10C) was not affected. Although both the *IR56d*[GAL4] mutants and the *IR56d*-silenced flies have reduced response to medium-chain fatty acids, the response of the *IR56d* mutant flies was relatively lower, likely caused by the complete deletion of the *IR56d* gene. Flies that were heterozygous for the *IR56d* deletion (*IR56d*[GAL4]/+) exhibited similar responses to those of control flies for all tastants measured. The observed decrease in PER to medium-chain fatty acids was rescued by transgenic expression of *IR56d* in the *IR56d*[GAL4] mutant background (*IR56d*[GAL4];UAS-*IR56d*/+), confirming that the behavioral deficit of *IR56d*[GAL4] flies is in fact due to loss of *IR56d* function. Similar results were observed in males. Although taste responses were generally lower, PER to sucrose did not differ between *IR56d*[GAL4] and control flies, but PER to medium-chain hexanoic acid (6C) was significantly lower (*Figure 4—figure supplement 1*). Therefore, *IR56d* appears to be selectively required for taste sensing of medium-chain fatty acids.

In previous work, we found that *IR56d*-expressing neurons are responsive to both sucrose and hexanoic acid (*Tauber et al., 2017*). To determine whether other classes of fatty acids can also evoke activity in these neurons and whether their activity is dependent on *IR56d*, we measured $Ca^{2+}$ responses to a panel of tastants. We expressed the $Ca^{2+}$ sensor GCaMP6.0 under the control of *IR56d*[GAL4] and measured tastant-evoked activity in the posterior projections (*Figure 5A–D*, *Figure 5—figure supplement 1*), which emanate from labellar taste neurons and are both necessary and sufficient for taste perception to medium-chain hexanoic acid (6C; *Koh et al., 2014*; *Tauber et al., 2017*). In flies heterozygous for *IR56d*[GAL4], the labeled neurons were responsive to sucrose and all fatty acids tested, which ranged from 4C to 10C (*Figure 5E*). Thus, *IR56d* neurons

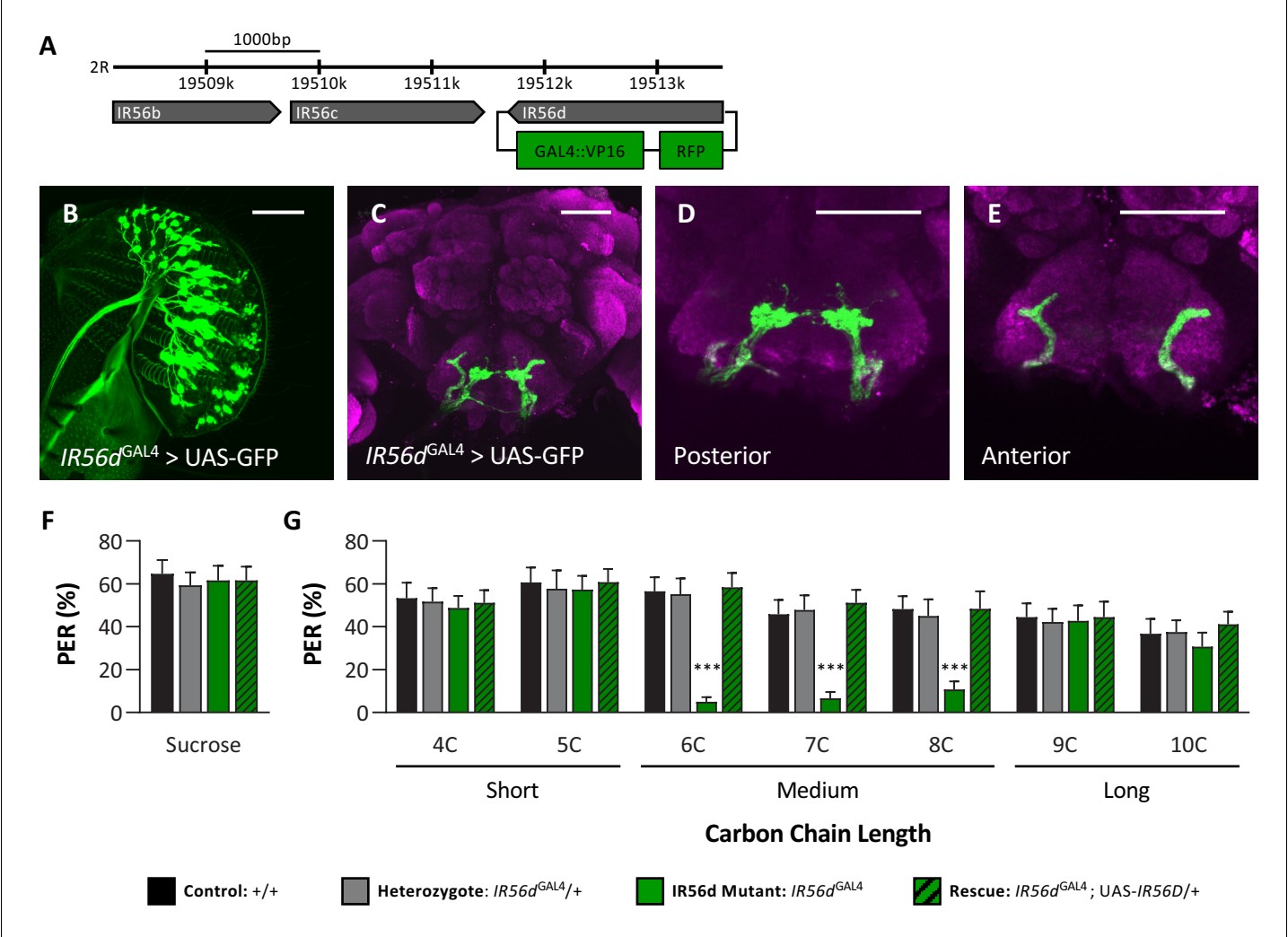

**Figure 4.** *IR56D* mediates taste perception to medium-chain fatty acids. (A) *IR56d*$^{GAL4}$ was generated using the CRISPR/Cas9 system. In *IR56d*$^{GAL4}$ flies, the *IR56d* gene was replaced by GAL4 and RFP elements (green boxes). The relative location and orientation of genes in the region are represented as gray arrows. (B–E) Expression pattern of *IR56d*$^{GAL4}$ is visualized with GFP. *IR56d*-expressing neurons are located on the (B) labellum and project to the (C) subesophageal zone of the brain. Distinct regions of projection include the (D) posterior and (E) anterior subesophageal zones. Background staining is NC82 antibody (magenta). Scale bar = 50 μm. (F) Sucrose taste perception is similar in control and *IR56d*$^{GAL4}$ mutant flies. Kruskal–Wallis test: H = 0.1758, p=0.9814, with Dunn's test for multiple comparisons; N = 33–40. (G) The *IR56d*$^{GAL4}$ flies have reduced proboscis extension response to medium-chain fatty acids (6C–8C) relative to control, *IR56d*$^{GAL4}$ heterozygotes, and *IR56d*$^{GAL4}$ rescue flies. However, all genotypes respond similarly to both short- and long-chain fatty acids (4C, 5C; 9C, 10C). Restricted maximum likelihood: F$_{3,850}$ = 17.80, p<0.0001, with Sidak's test for multiple comparisons; N = 28–40. Error bars indicate ± SEM. ****p<0.0001.

The online version of this article includes the following source data and figure supplement(s) for figure 4:

**Source data 1.** Raw data from IR56d$^{GAL4}$ proboscis extension response experiments.

**Figure supplement 1.** *IR56d* mediates taste perception to medium-chain hexanoic acid (6C) in male flies.

respond to diverse appetitive stimuli. Flies with a deletion of *IR56d* (*IR56d*$^{GAL4}$; *UAS*-GCaMP6.0) lacked responses exclusively to medium-chain fatty acids (6C–8C), while responses to short- (4C and 5C) and long-chain fatty acids (9C and 10C) remained intact (*Figure 5F*). Consistent with the rescue of behavioral defects, inclusion of an *IR56d* rescue transgene (*IR56d*$^{GAL4}$; *UAS*-GCaMP6.0/*UAS*-*IR56d*) restored the physiological response to medium-chain fatty acids (*Figure 5G*). Quantification of the responses to all tastants confirmed that Ca$^{2+}$ responses to 6C–8C fatty acids are disrupted in *IR56d*$^{GAL4}$ flies and restored to levels observed in control flies by expression of UAS-*IR56d* (*Figure 5H*). Overall, these results demonstrate that at both behavioral and physiological levels *IR56d*$^{GAL4}$ is required for taste responses to medium-chain fatty acids.

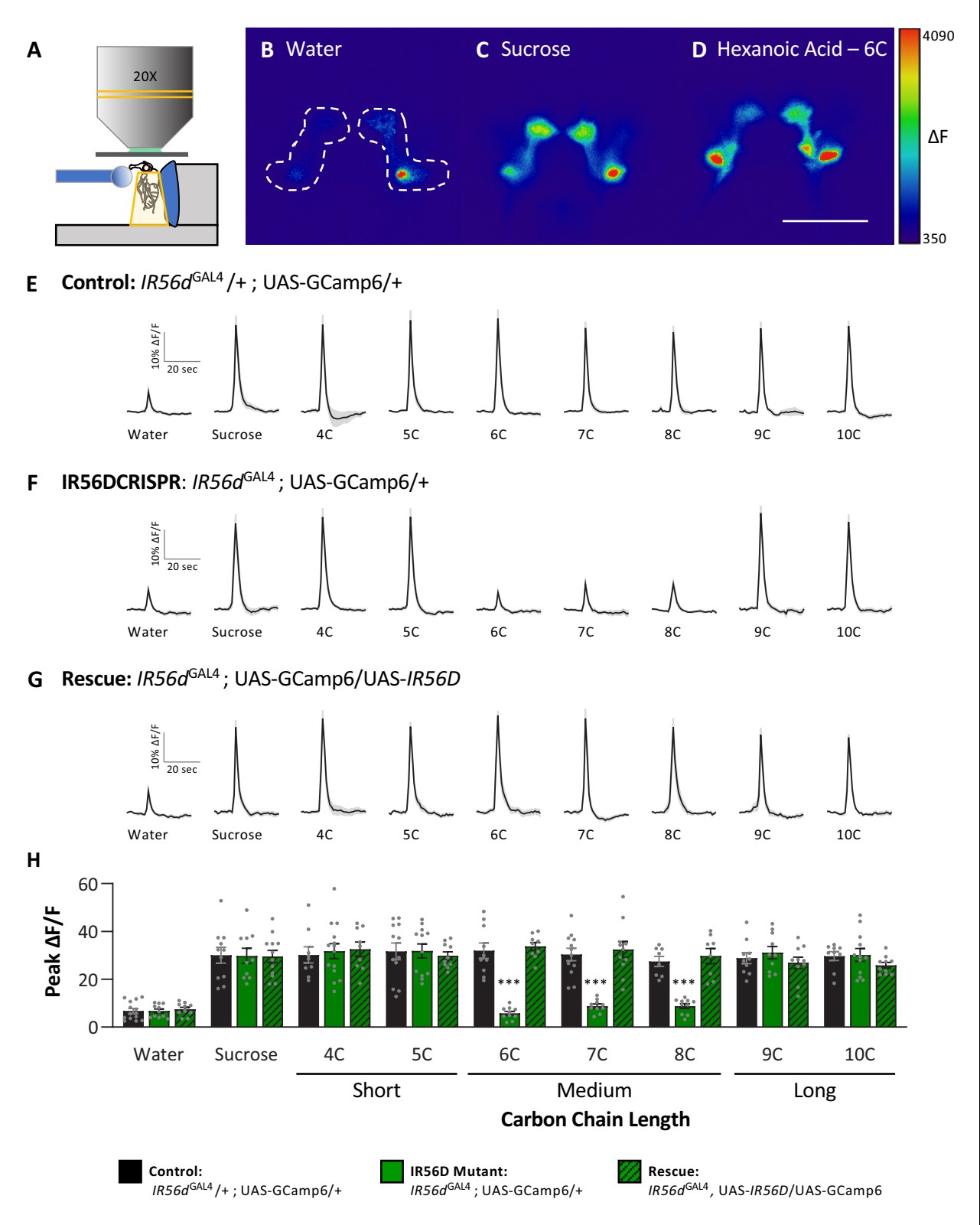

**Figure 5.** Neuronal responsiveness in posterior labellar taste neurons of *IR56d*^GAL4 mutant flies is reduced in response to medium-chain fatty acids. (**A**) Diagram of live-imaging experimental protocol. A tastant is applied to the proboscis while florescence is recorded simultaneously. (**B–D**) Representative pseudocolor images of calcium activity in the posterior projections of *IR56D* neurons in response to water (**B**), 10 mM sucrose (**C**), or 1% hexanoic acid (**D**). Shown is the change in UAS-GCaMP6 fluorescence (ΔF). Scale bar = 50 μm. (**E–G**) Activity traces of the posterior projections of *IR56D* neurons in

*Figure 5 continued on next page*

*Figure 5 continued*

response to each tastant in the (E) IR56d<sup>GAL4</sup> heterozygote controls, (F) IR56d<sup>GAL4</sup> mutants, and (G) IR56d<sup>GAL4</sup> rescue flies. The shaded region of each trace indicates ± SEM. (H) Average peak change in fluorescence for data shown in (E–G). Neuronal responses to medium-chain fatty acids (6C–8C) are significantly reduced in IR56d<sup>GAL4</sup> mutants compared to IR56d<sup>GAL4</sup> heterozygote controls and IR56d<sup>GAL4</sup> rescue flies. All genotypes respond similarly to both short- and long-chain fatty acids (4C, 5C; 9C, 10C), as well as to water and sucrose. Two-way ANOVA: $F_{2,256} = 23.67$, $p<0.0001$, with Sidak's test for multiple comparisons; N = 8–14. Error bars indicate ± SEM. ***$p<0.001$.

The online version of this article includes the following source data and figure supplement(s) for figure 5:

**Source data 1.** Raw imaging data from the posterior labellar region in IR56d<sup>GAL4</sup> flies.

**Figure supplement 1.** Representative pseudocolor images of calcium activity in the posterior projections of *IR56d* neurons in response to fatty acid presentation.

The neurons expressing *IR56d* project to two distinct regions within the SEZ: a posterior set of projections that overlap with *GR64f*-expressing neurons and a set of anterior projections that originate from the taste pegs (*Koh et al., 2014*; *Tauber et al., 2017*). The two populations appear to differ in function as the posterior population is responsive to both sucrose and hexanoic acid, while the anterior population is responsive to hexanoic acid, but not sucrose (*Tauber et al., 2017*). It is possible that the anterior neurons are selectively responsive to medium-chain fatty acids, providing a mechanism for taste discrimination. We generated flies labeling *IR56d*- and *GR64f*-expressing neurons with different genetically encoded fluorescent reporters (IR56d<sup>GAL4</sup>; UAS-RFP; GR64f-LexA > LexAop GFP) and confirmed that IR56d<sup>GAL4</sup> labels both the anterior and posterior projections (*Figure 6A–C*). To selectively measure fatty acid responses in anterior taste peg neurons, we expressed *UAS-GCaMP6* with a genetic intersection strategy (IR56d<sup>GAL4/+</sup>; UAS-GCaMP6; GR64f-LexA > LexAop-GAL80) and measured activity to a panel of tastants. In agreement with our previous findings, these neurons were robustly responsive to hexanoic acid (6C), but not by sucrose (*Figure 6D–F*; *Tauber et al., 2017*). These neurons also responded to other classes of fatty acids including short- (4C and 5C), medium- (7C and 8C) and long-chain (9C and 10C) (*Figure 6D–F*). Unlike the posterior projections, which respond similarly to all three classes of fatty acids, the response to short-chain fatty acids was lower in the anterior population. Therefore, it is possible that differential response between the anterior and posterior populations provides a mechanism of discrimination between different classes of fatty acids. However, this alone is insufficient to fully explain discrimination between medium- and long-chain fatty acids. Taken together, these findings support the notion that medium-chain fatty acids are detected through a shared sensory channel, allowing flies to distinguish medium-chain from short- or long-chain, but not between different medium-chain fatty acids.

## Discussion

Receptors for sweet and bitter taste have been well defined in both flies and mammals (*Carleton et al., 2010*; *Hallem et al., 2006*; *Scott, 2018*), but less is known about detection of fats. Previous studies identified *IR56d* as a receptor for hexanoic acid and carbonation (*Ahn et al., 2017*; *Sánchez-Alcañiz et al., 2018*). Our findings suggest that *IR56d* is selectively involved in responses to medium-chain fatty acids, including 6C, 7C, and 8C fatty acids, and dispensable for responses to shorter and longer-chain fatty acids. Such receptor specificity for different classes of fatty acids based on chain length has not been documented in other systems. In flies, both sugars and fatty acids evoke activity in neurons that co-express the receptors *GR64f* and *IR56d*. The finding that short- and long-chain fatty acids also evoke activity in *IR56d*-expressing neurons posits that additional fatty acid receptors are present in these neurons. Previously, we and others have found that deletion of Phospholipase C (PLC) signaling selectively impairs fatty acid response while leaving sweet taste intact, raising the possibility that activation of distinct intracellular signaling pathways could serve as a mechanism for discrimination between sucrose and fatty acid (*Masek and Keene, 2013*; *Ahn et al., 2017*; *Tauber et al., 2017*), while another suggests *TRPA1* and *GR64e* are targets of PLC and are generally required for fatty acid sensing (*Kim et al., 2018*). Determining whether or not short- and long-chain fatty acids also signal through PLC may provide insight into whether signaling mechanisms are shared between different fatty acid receptors expressed in *IR56d*-expressing neurons.

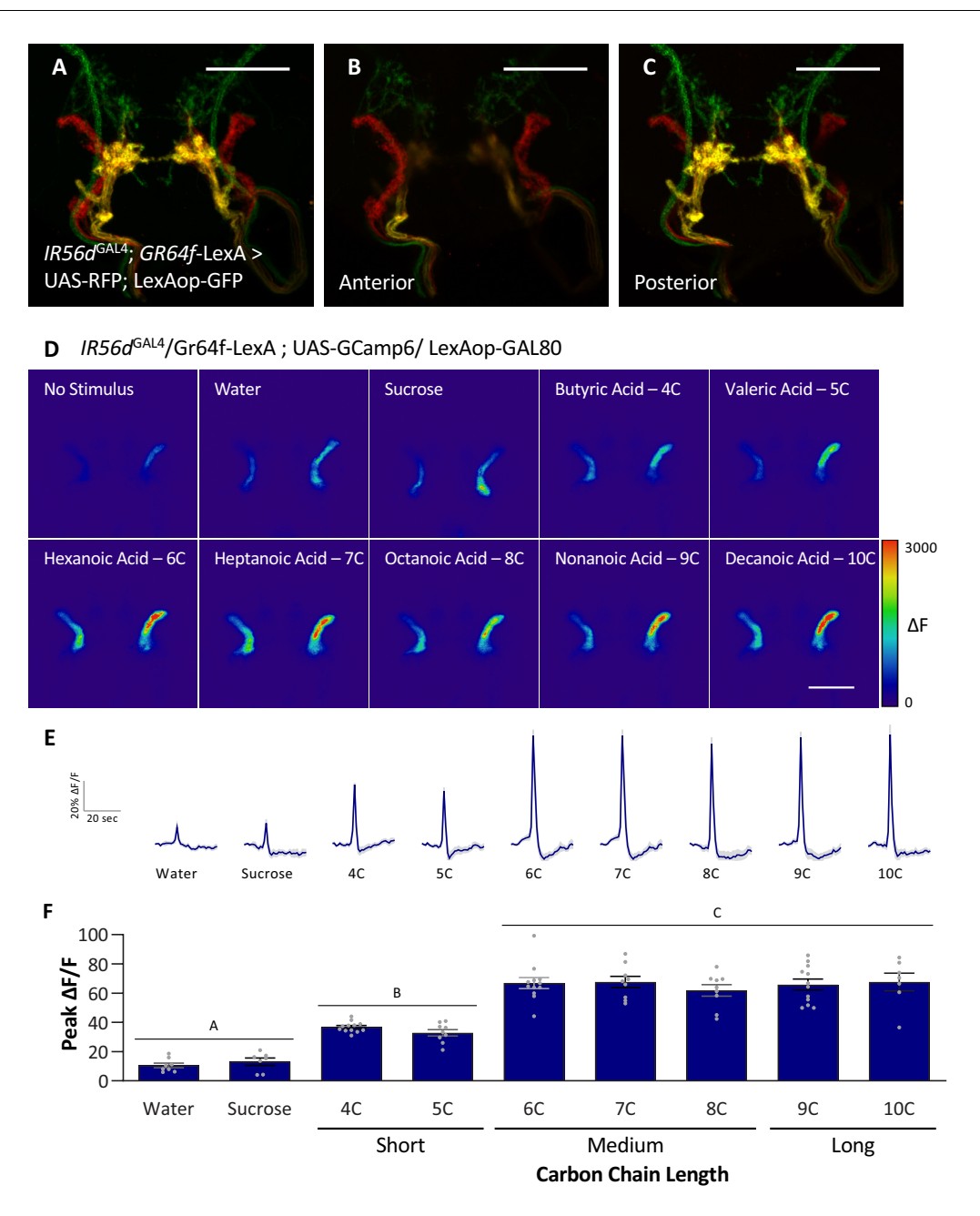

**Figure 6.** Anterior, non-*GR64f*-*IR56d*-expressing neurons are response to short-, medium-, and long-chain fatty acids. (A–C) Colocalization of *IR56d*<sup>GAL4</sup> and *GR64f*-expressing neurons occurs in the posterior subesophageal zone. (A) Expression pattern of *IR56d*<sup>GAL4</sup> and *GR64f* is visualized with *IR56d*<sup>GAL4</sup> driving UAS-RFP and *GR64f*-LexA driving LexAop-GFP. Colocalization (yellow) is detected in the posterior projections (B), but not in the anterior projections (C). Scale bar = 50 µm. (D–F) Restricting UAS-GCaMP6 expression to the non-overlapping anterior projection neurons does not significantly impact neuronal activity in response to short-, medium-, or long-chain fatty acid presentation. Live imaging was performed as described in *Figure 5A*. (D) Representative pseudocolor images of calcium activity in the anterior projections of non-*GR64f*-*IR56d*-expressing neurons in response to tastant presentation. Shown is the change in UAS-GCaMP6 fluorescence (ΔF). Scale bar = 50 µm. (E) Activity traces of the anterior projections of non-*GR64f*-*IR56d*-expressing neurons in response to each tastant in the *IR56d*<sup>GAL4</sup> rescue flies. The shaded region of each trace indicates ± SEM. (F) Average peak change in fluorescence for data shown in (E). Neuronal activity in response to water and sucrose presentation is significantly reduced compared to fatty acid presentation, while responses to short-chain fatty acids (4C, 5C) are intermediate compared to medium- and long-chain fatty acids (6C–10C). No difference in neural activity between medium- and long-chain fatty acid presentation was observed. One-way ANOVA: $F_{8,76} = 45.22$, p<0.0001, with Sidak's test for multiple comparisons; N = 7–12. Error bars indicate ± SEM. ***p<0.001.

The online version of this article includes the following source data for figure 6:

**Source data 1.** Raw imaging data from the anterior, non-*GR64f*-*IR56d*-expressing region in IR56d<sup>GAL4</sup> flies.

Our aversive taste memory assay confirmed previous findings that flies can discriminate between sugars and fatty acids (*Tauber et al., 2017*), and led to the surprising observation that flies can distinguish between different classes of fatty acids, even though the baseline responsiveness to short-, medium-, and long-chain fatty acids was similar in innate preference assays. Fatty acids are natural by-product of yeast fermentation (*Diwan and Gupta, 2018*; *Nyanga et al., 2013*; *Oliveira et al., 2011*), and their abundance in peaches, for example, declines after ripening (*Duan et al., 2013*). Further, fatty acids have antifungal activity, which scales with chain length (i.e., the greater the chain length, the greater the antifungal efficiency; *Pohl et al., 2011*). Thus, the ability to discriminate between different classes of fatty acids is likely to be important in determining the stage of fruit ripeness, degree of fermentation, and the general palatability of a potential food source/oviposition site.

The finding that flies can distinguish between different classes of fatty acids contrasts with the results of a previous study that applied a similar assay and found that flies were unable to discriminate between different sugars or bitter compounds (*Masek and Scott, 2010*). One possibility is that this is due to differences in fatty acid detection, which is dependent on IRs, and sweet and bitter tastant detection, which relies on GRs (*Chen and Dahanukar, 2020*). Our findings highlight the complexity of taste discrimination, which extends beyond simple PER as a readout for taste. For example, all types of fatty acids tested increase GR64f neural responsiveness; however, only GR64f neurons are required for PER to medium-chain fatty acids, thereby raising the possibility that short-, medium-, and long-chain fatty acid taste discrimination occurs through different neural channels. These findings stress the need to define the fatty acid receptors and neural circuits that govern responses to short- and long-chain fatty acid taste. Furthermore, the ability of the *Drosophila* the taste system to discriminate suggests it may be more like the olfactory system than previously appreciated. Flies are able to distinguish between many different odorants, likely due to the complexity of olfactory coding at the level of the receptor as well as in the antennal lobe (*Amin and Lin, 2019*; *Cognigni et al., 2018*; *Guven-Ozkan and Davis, 2014*). However, flies can also discriminate between odorants sensed by a single olfactory receptor, suggesting that temporal coding also plays a role in discrimination (*DasGupta and Waddell, 2008*). It is possible that similar mechanisms underlie discrimination between different classes of fatty acid tastants.

The *Drosophila* genome encodes 66 IRs, which comprise a recently identified family of receptors implicated in taste, olfaction, and temperature sensation (*Benton et al., 2009*; *Rytz et al., 2013*). IRs are involved in the detection of many different tastants and function as heteromers that confer sensory specificity (*Rytz et al., 2013*; *van Giesen and Garrity, 2017*). While *IR56d* expression is restricted to a subset of sweet taste neurons, it likely functions in a complex with *IR25a* and *IR76b*, all three of which are required fatty acid taste (*Ahn et al., 2017*; *Sánchez-Alcañiz et al., 2018*). Other tastants whose responses are mediated by IRs are also likely to be detected by IR complexes. For example, roles for *IR25a*, *IR62a*, and *IR76b* have been described for $Ca^{2+}$ taste (*Lee et al., 2018*). The broad degree of co-expression of IRs in the brain and periphery can provide candidates for those involved in detecting short- and long-chain fatty acids.

The identification of taste discrimination between different classes of fatty acids provides the opportunity to identify how different tastants are encoded in the brain and how these circuits are modified with experience. Although projections of primary taste neurons to the SEZ have been mapped in some detail, little is known about connectivity with downstream neurons and whether sensory neurons responsive to different appetitive tastants can activate different downstream circuits. Recent studies have identified a number of interneurons that modify feeding, including IN1, a cholinergic interneuron responsive to sucrose (*Yapici et al., 2016*), E564 neurons that inhibit feeding (*Mann et al., 2013*), and *Fdg* neurons that are required for sucrose-induced feeding (*Flood et al., 2013*). Future work can investigate whether these and other downstream neurons are shared for fatty acid taste. Previous studies have found that incoming sensory information is selectively modulated within the SEZ in accordance with feeding state (*Chu et al., 2014*; *LeDue et al., 2016*). It will be interesting to determine if similar modulation promotes differentiation of sugars and fatty acids, which are sensed by shared GRNs. Large-scale brain imaging has now been applied in flies to measure responsiveness to different tastants (*Harris et al., 2015*), and a comparison of brain activity patterns elicited by different classes of fatty acids may provide insight into differences in their sensory input and processing.

All experiments in this study tested flies under starved conditions, which is necessary to elicit the PER that is used as a behavioral readout of taste acceptance. However, responses to many tastants and odorants are altered in accordance with feeding state (*LeDue et al., 2016*; *Root et al., 2008*). For example, the taste of acetic acid is aversive to fed flies but attractive to starved flies, revealing a hunger-dependent switch (*Devineni et al., 2019*). Similarly, hexanoic acid evokes activity in both sweet and bitter-sensing taste neurons, and the activity of bitter taste neurons is dependent on different receptors from those involved in the appetitive response (*Ahn et al., 2017*). Further, hunger enhances activity in sweet taste circuits and suppresses that of bitter taste circuits, providing a mechanism for complex state-dependent modulation of taste response that increase activity of both appetitive and deterrent neurons (*Inagaki et al., 2014*; *LeDue et al., 2016*).

The neural circuits that are required for aversive taste memory have been well defined for sugar, yet little is known about how fatty acid taste is conditioned. The pairing of sugar with bitter quinine results in aversive memory to sugar. Optogenetic activation of bitter taste neurons that are activated by quinine, in combination with the presentation of sugar, is sufficient to induce sugar avoidance (*Keene and Masek, 2012*). Further studies have elucidated that aversive taste memories are dependent on mushroom body neurons that form the gamma and alpha lobes, the PPL1 cluster of dopamine neurons, and alpha lobe output neurons, revealing a circuit regulating taste memory that differs from that controlling appetitive olfactory memory (*Kirkhart and Scott, 2015*; *Masek et al., 2015*). It will be interesting to determine whether shared components regulate conditioning to fatty acids or whether distinct mushroom body circuits regulate sweet and fatty acid taste conditioning. Further, examination of the central brain circuits that regulate aversive taste conditioning to different classes of fatty acids will provide insight into how taste discrimination is processed within the brain.

## Materials and methods

### *Drosophila* stocks and maintenance

Flies were grown and maintained on standard food media (Bloomington Recipe, Genesee Scientific, San Diego, CA). Flies were housed in incubators (Powers Scientific, Warminster, PA) on a 12:12 LD cycle at 25°C with humidity of 55–65%. The following fly strains were ordered from the Bloomington Stock Center: $w^{1118}$ (#5905; *Levis et al., 1985*), IR56d-GAL4 (#60708; *Koh et al., 2014*), UAS-impTNT (#28840; *Sweeney et al., 1995*), UAS-TNT (#28838; *Sweeney et al., 1995*), $IR25a^1$ (#41376; *Benton et al., 2009*); $IR76b^2$ (#51310; *Zhang et al., 2013*), UAS-GFP (#32186; *Pfeiffer et al., 2010*), UAS-GCaMP5 (#42037; *Akerboom et al., 2012*), UAS-RFP;GFP-LexAop (#32229; *Pfeiffer et al., 2010*), and LexAop-GAL80 (#32213). *GR64f*-LexA was kindly provided by H. Tanimoto and was previously described in *Thoma et al., 2016*. UAS-*IR56d* was generated using *IR56d* cDNA, amplified with primers that generated a NotI-KpnI fragment that was cloned in the pUAS vector. The *IR56d*-$^{GAL4}$ line was generated by WellGenetics (Taipei City, Taiwan) using the CRISPR/Cas9 system to induce homology-dependent repair. At the gRNA target site, a dsDNA donor plasmid was inserted containing a GAL4::VP16 and RFP cassette. This line was generated in the $w^{1118}$ genetic background and was validated by PCR and sequencing. All lines were backcrossed to the $w^{1118}$ fly strain for 10 generations. Unless stated otherwise, mated female flies aged 7–9 days were used. For ablation experiments, the antenna and/or maxillary palp were removed 2 days post eclosion.

### Reagents

The following fatty acids were obtained from Sigma Aldrich (St. Louis, MO): butyric acid (4C; #B103500), valeric acid (5C; #240370), hexanoic acid (6C; #21530), heptanoic acid (7C; #75190), octanoic acid (8C; #O3907), nonanoic acid (9C; #N5502), and decanoic acid (10C; #C1875). All fatty acids were dissolved in water, although both nonanoic and decanoic acids required heating to fully go into solution. Quinine hydrochloride was also obtained from Sigma Aldrich (#Q1125), while sucrose was purchased from Fisher Scientific (#FS S5-500; Hampton, New Hampshire).

### Immunohistochemistry

Brains were prepared as previously described (*Kubrak et al., 2016*). Briefly, brains of 7–9-day-old female flies were dissected in ice-cold phosphate buffered saline (PBS) and fixed in 4% formaldehyde, PBS, and 0.5% Triton-X for 30 min at room temperature. Brains were rinsed 3× with PBS and

0.5% Triton-X (PBST) for 10 min at room temperature and then incubated overnight at 4°C. The next day brains were incubated in primary antibody (1:20 mouse nc82; Iowa Hybridoma Bank; The Developmental Studies Hybridoma Bank, Iowa City, IA) diluted in 0.5% PBST at 4°C for 48 hr. Next, the brains were rinsed 3× in 0.5% PBST 3 × 10 min at room temperature and placed in secondary antibody (1:400 donkey anti-mouse Alexa Fluor 647; #A-31571; ThermoFisher Scientific, Waltham, MA) for 90 min at room temperature. The brains were again rinsed 3× in PBST for 10 min at room temperature and then mounted in Vectashield (VECTOR Laboratories, Burlingame, CA). Brains were imaged in 2 µm sections on a Nikon A1R confocal microscope (Nikon, Tokyo, Japan) using a 20× oil immersion objective. Images presented as the Z-stack projection through the entire brain and processes using ImageJ2 (*Tauber et al., 2017*).

## Proboscis extension response

Female flies were starved for 24 hr prior to each experiment and then PER was measured as previously described (*Masek and Keene, 2013*; *Tauber et al., 2017*). In experiments using males, flies were starved for 18 hr prior to each experiment. Briefly, flies were anesthetized on $CO_2$ and then restrained inside of a cut 200 µL pipette tip (#02-404-423; Fisher Scientific) so that their head and proboscis were exposed while their body and tarsi remain restrained. After a 60 min acclimation period in a humidified box, flies were presented with water and allowed to drink freely until satiated. Flies that did not stop responding to water within 5 min were discarded. A wick made of Kimwipe (#06-666; Fisher Scientific) was placed partially inside a capillary tube (#1B120F-4; World Precision Instruments, Sarasota, FL) and then saturated with tastant, thereby enabling flies to taste, but not ingest tastant. The saturated wick was then manually applied to the tip of the proboscis for 1–2 s and proboscis extension reflex was monitored. Only full extensions were counted as a positive response. Each tastant was presented a total of three times, with 1 min between each presentation. Unless otherwise stated, fatty acids were dissolved in water and tested at a concentration of 1%, while sucrose was tested at a concentration of 10 mM. PER was calculated as the percentage of proboscis extensions divided by the total number of tastant presentations. For example, a fly that extends its proboscis twice out of the three presentations will have a PER response of 66%. Experiments were run blinded, ~3 times per week until completion. Additionally, genotype and tastant presentation was randomized to ensure data reproducibility.

## In vivo calcium imaging

Female flies were starved for 24 hr prior to imaging, as described (*Tauber et al., 2017*). Flies were anesthetized on ice and then restrained inside of a cut 200 µL pipette tip so that their head and proboscis were accessible, while their body and tarsi remain restrained. The proboscis was manually extended and then a small amount of dental glue (#595953WW; Ivoclar Vivadent Inc, Amherst, NY) was applied between the labium and the side of the pipette tip, ensuring the same position throughout the experiment. Next, both antennae were removed. A small hole was cut into a 1 $cm^2$ piece of aluminum foil and then fixed to the fly using dental glue, creating a sealed window of cuticle exposed. Artificial hemolymph (140 mM NaCl, 2 mM KCl, 4.5 mM $MgCl_2$, 1.5 mM $CaCl_2$, and 5 mM HEPES-NaOH with pH = 7.1) was applied to the window and then the cuticle and connective tissue were dissected to expose the SEZ. Mounted flies were placed on a Nikon A1R confocal microscope and then imaged using a 20× water-dipping objective lens. The pinhole was opened to allow a thicker optical section to be monitored. All recordings were taken at 4 Hz with 256 resolution. Similar to PER, tastants were applied to the proboscis for 1–2 s with a wick, which was operated using a micromanipulator (Narishige International USA, Inc, Amityville, NY). Experiments were run ~3 times per week until completion. For analysis, regions of interest were drawn manually around posterior *IR56D* projections. Baseline fluorescence was calculated as the average fluorescence of the first five frames, beginning 10 s prior to tastant application. For each frame, the % change in fluorescence (%ΔF/F) was calculated as: (peak fluorescence – baseline fluorescence)/baseline fluorescence * 100. Average fluorescence traces were created by taking the average and standard error of %ΔF/F for each recording of a specific tastant.

## Aversive taste memory

Taste discrimination was assessed by measuring aversive taste memory, as described previously (*Tauber et al., 2017*). Female flies were starved for 24 hr prior to each experiment. Flies were then anesthetized on $CO_2$ and the thorax of each fly was glued to a microscope slide using clear nail polish (#451D; Wet n Wild, Los Angeles, CA). Flies were acclimated to these conditions in a humidified box for 60 min. For each experiment, the microscope slide was mounted vertically under a dissecting microscope (#SM-1BSZ-144S; AmScope, Irvine, CA). Flies were water satiated prior to each experiment and in between each test/training session. For tastant presentation, we used a 200 µL pipette tip attached to a 3 mL syringe (#14-955-457; Fisher Scientific). For the pretest, 1% fatty acid was presented to the proboscis three times with a 30 s interval between applications, and the number of full proboscis extensions was recorded. During training, a similar protocol was used except that each tastant presentation was immediately followed by 50 mM quinine presentation, in which flies were allowed to drink for up to 2 s or until an extended proboscis was retracted. A total of three training sessions were performed. In between each session, the proboscis was washed with water and flies were allowed to drink to satiation, lasting ~2 min. To assess taste discrimination, flies were tested either with that same tastant without quinine or with an untrained tastant. Another group of flies were tested as described above but quinine was never presented (naïve). At the end of each experiment, flies were given 1 M sucrose to check for retained ability to extend proboscis and all non-responders were excluded.

## Statistical analysis

All measurements are presented as bar graphs showing mean ± standard error. Measurements of PER and aversive taste memory were not normally distributed and so the non-parametric Kruskal–Wallis test was used to compare two or more genotypes. To compare two or more genotypes and two treatments, a restricted maximum likelihood (REML) estimation was used. For data that was normally distributed (calcium imaging data), a one-way or two-way analysis of variance (ANOVA) was used for comparisons between two or more genotypes and one treatment or two or more genotypes and multiple treatments, respectively. All post hoc analyses were performed using Sidak's multiple comparisons test. Statistical analyses and data presentation were performed using InStat software (GraphPad Software 8.0, San Diego, CA). Sample sizes for behavioral and functional imaging experiments are consistent with previous studies (*Kirkhart and Scott, 2015*; *Tauber et al., 2017*). Generally, ~30 flies were used for each experimental or control group for behavioral experiments and ~10–12 flies per group for imaging experiments.

## Acknowledgements

We would like to thank members of the Keene and Dahanukar labs for technical assistance and helpful discussion. This work was supported by NIH grants R01 NS085252 to ACK and R01DC017390 to ACK and AD, as well as support from FAU's Jupiter Life Science Initiative.

## Additional information

### Funding

| Funder | Grant reference number | Author |
| --- | --- | --- |
| National Institutes of Health | NIH R01DC017390 | Elizabeth B Brown<br>Kreesha D Shah<br>Justin Palermo<br>Manali Dey<br>Anupama Dahanukar |

The funders had no role in study design, data collection and interpretation, or the decision to submit the work for publication.

## Author contributions
Elizabeth B Brown, Conceptualization, Data curation, Formal analysis, Investigation, Visualization, Methodology, Writing - original draft, Writing - review and editing; Kreesha D Shah, Formal analysis, Validation, Investigation, Writing - original draft; Justin Palermo, Formal analysis, Validation, Investigation, Visualization, Writing - review and editing; Manali Dey, Resources, Formal analysis, Investigation, Writing - original draft, Writing - review and editing; Anupama Dahanukar, Conceptualization, Formal analysis, Investigation, Visualization, Writing - original draft, Project administration, Writing - review and editing; Alex C Keene, Conceptualization, Resources, Supervision, Writing - original draft, Project administration, Writing - review and editing

## Author ORCIDs
Elizabeth B Brown (ID) https://orcid.org/0000-0003-0825-8199
Alex C Keene (ID) https://orcid.org/0000-0001-6118-5537

## Decision letter and Author response
Decision letter https://doi.org/10.7554/eLife.67878.sa1
Author response https://doi.org/10.7554/eLife.67878.sa2

---

# Additional files

## Supplementary files
• Transparent reporting form

## Data availability
All data generated or analysed during this study are included in the manuscript and supporting files. Source data files have been provided for all figures.

---

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
