## [Decision Letter]

**Acceptance summary:**

In this manuscript, the authors build on their previous work on fatty acid detection by the taste system of *Drosophila melanogaster*. They show, by developing a novel behavioral assay, that flies can distinguish medium-length fatty acids from other fatty acids by using different taste sensilla. These findings suggest that flies have the capacity to discriminate molecules of the same chemical class.

**Decision letter after peer review:**

[Editors’ note: the authors submitted for reconsideration following the decision after peer review. What follows is the decision letter after the first round of review.]

Thank you for submitting your work entitled "Ir56d-dependent fatty acid responses in *Drosophila* uncover taste discrimination between different classes of fatty acids" for consideration by *eLife*. Your article has been reviewed by 3 peer reviewers, and the evaluation has been overseen by Kristin Scott as Reviewing Editor and a Senior Editor. The reviewers have opted to remain anonymous.

Our decision has been reached after consultation between the reviewers. Based on these discussions and the individual reviews below, we regret to inform you that your work will not be considered further for publication in *eLife* in its current form.

The reviewers find fatty acid taste discrimination potentially interesting and agree that the experiments are performed to a high standard. One major concern is whether discrimination is based on intensity rather than quality. A second limitation is that the mechanism of FA detection is not greatly advanced beyond the authors' previous work: the cellular mechanisms for long and short chain FA detection remain unclear. The reviewers agreed that if the major concerns of Reviewer 1 were addressed, this manuscript would provide a broader understanding of fatty acid discrimination.

*Reviewer #1:*

This paper investigates fatty acid taste in flies and asks the broad question of whether flies can discriminate different compounds within a single taste modality. The authors' main finding is that flies can discriminate between long, medium, and short chain fatty acids using a previously established aversive memory taste paradigm. When they delve into the cellular and molecular basis of fatty acid detection they find that IR56d neurons respond to all three classes of fatty acids, but are required only for the behavioural responses to medium chain molecules. Similarly, CRISPR/Cas9 deletion of the IR56d receptor reveals that it too is required only for medium-chain fatty acid responses. Thus, different fatty acid classes presumably activate distinct, but partially overlapping subsets of appetitive taste neurons. In general I think the paper is potentially interesting (see comment 1 below) and the data mostly supports the conclusions. However, there is some lack of attention to details that make some of the data hard to interpret.

1. The ability of flies to discriminate between different fatty acid classes is presented as the interesting finding, since, as the authors point out, discrimination between compounds within a taste modality is generally not thought to occur. On the surface I agree that this is interesting. However, in the authors' set up of the main question (line 101), they raise an important issue: "Is it possible that flies are capable of differentiating between tastants of the same modality, or is discrimination within a modality exclusively dependent on concentration?" This should be rephrased to replace "concentration" with "intensity" since not all tastants at the same concentration have the same intensity, and from a behavioural perspective it is intensity that matters. Given that, the authors don't do anything to demonstrate that their discrimination task does not depend on intensity, aside from the fact that 1% solutions of all the FA seem to give similar PER. They need to show more explicitly that this task is truly showing identity-based discrimination.

2. The second broad concern I have is over the nature of short and long chain fatty acid detection. Interpreting the discrimination results would be greatly aided if we knew what other neurons mediate the PER to these molecules. Is it the non-IR56d population of Gr64f neurons? Two experiments would go a long way to addressing this question: (1) silence Gr64f neurons to test whether the broader population is required for short and long chain FA PER and (2) do calcium imaging of Gr64f neurons and see whether non-IR56d projections (which according to the authors' previous work are spatially segregated in a more dorsal area of the SEZ) respond to short and long chain FA.

*Reviewer #2:*

In the present paper Brown et al., study the ability of *Drosophila melanogaster* to discriminate between Fatty Acids (FAs) of different lengths. Using a combination of behavioral experiments, molecular biology and in vivo calcium imaging, the authors show that a subset of Ir56d expressing neurons are able to differentiate FAs. However, the Ir56d receptor is only necessary for the detection of medium-length FAs but not short- or long-. The paper explores in detail the role of the Ir56d receptor as FA detector, a role previously described by the authors in a previous paper Tauber et al. 2017.

I consider that the experiments are properly done, and so the statistical analysis, however gain in knowledge is very limited. So far, the authors can prove that flies can discriminate FAs of different lengths, being Ir56d the receptor detecting medium-length FAs, a result that expands the knowledge gained in Tauber et al. 2017. In figure 3, the authors show that silencing Ir56d neurons using tetanus toxin expression, reduces dramatically PER to medium-length fatty acids, but not to short or long, pointing to a different set of neurons involved in their detection. However, the in vivo calcium imaging experiments show that Ir56d neurons also respond to short- and long- FAs. In this regard, I disagree with the statement at the abstract: Characterization of hexanoic acid-sensitive Ionotropic receptor 56d (Ir56d) neurons reveals broad responsive to short-, medium-, and long- chain fatty acids, suggesting selectivity is unlikely to occur through activation of distinct sensory neuron populations. In fact, I consider that selectivity would come from the activation of different subsets of gustatory neurons. It seems that Ir56d neurons could be a subset of the neurons that generally respond to FAs, providing the specificity for medium-length FAs. Other neurons, in addition to the Ir56d ones might be responding to short- and long- FAs in an Ir56d independent manner.

I consider the authors should explore in deep how short- and long- FAs are actually detected, whether it depends on other Ionotropic Receptors (probably Ir25a and Ir76b might be involved (Ahn et al. 2017)) and which subset of gustatory neurons are actually responding to these compounds, considering they do not require Ir56d nor Ir56d neurons.

*Reviewer #3:*

In this manuscript Brown et al. characterized fatty acid taste discrimination in *Drosophila melanogaster*. Fat taste is relatively poorly understood, but has critical implications for feeding and obesity research; thus, studies that advance our understanding of the molecular and physiological underpinning of this modality are important. The finding that Ir56d neurons enable organisms to discriminate between short, medium and long chain fatty acids but not to differentiate between types of medium chain fatty acids is certainly novel and interesting. It is also surprising but fascinating that this receptor is only required for the detection of medium fatty acids. The manuscript is well written and the figures presented in a clear and thoughtful manner. These findings lay out ground for future exciting work to investigate how sweet taste and fatty acid taste perception are selectively modulated by the brain since these gustatory neurons overlap and whether such discrimination is altered depending on the state of hunger.

1. Despite the overlapping nature of taste neurons in this case, i.e., Ir56d neurons being co-expressed with Gr64f – those that broadly label the sweet GRNs and the fact that Ir56d neurons are responsive to both sucrose and fatty acids; mutation in Ir56d results in loss of taste for hexanoic acid, but not sucrose. Authors use this taste discrimination to their advantage in combination with a robust aversive taste memory assay to address the question of differential fatty acid taste perception.

2. Authors rule out the potential involvement of olfaction in modulating taste perception.

3. Use of CRISPR-Cas9 to generate Ir56dGAL4 flies, implying accurate and targeted genome editing, provide validation to the results obtained when Ir56d expressing neurons are silenced. Additionally, use of the fly gustatory system for in-vivo Ca^2+^ imaging strengthens and corroborates the results at the physiological level, especially the rescue experiments.

Overall comments and questions:

1. Are the differences in taste discrimination between male and female flies?

2. Individual data points should be shown whenever possible for all figures (except PER because that would make it impossible to interpret).

3. Can the authors discuss how discriminating between different fatty acids types may be adaptive? Are they found in different food sources, some of which are "good" and some "bad"? Is there evidence from other organisms about this type of molecular discrimination in fatty acid taste?

[Editors’ note: further revisions were suggested prior to acceptance, as described below.]

Thank you for resubmitting your work entitled "Ir56d-dependent fatty acid responses in *Drosophila* uncovers taste discrimination between different fatty acid classes" for consideration by *eLife*.

Your revised article has been evaluated by a Senior Editor, a Reviewing Editor and three reviewers who wish to remain anonymous.

Essentially, the reviewers find your observation that flies can distinguish medium-length fatty acids from other fatty acids very interesting. They also praise the behavioral assay that you have developed to test this ability. Although the reviewers appreciate your revisions, unfortunately, the earlier concerns regarding a lack of concrete mechanism beyond what you have previously reported remain. Specifically, the reviewers felt that the molecular and circuit mechanisms underpinning the ability to discern medium chain fatty acids are still unresolved. Moreover, they would have liked to see some data elucidating how the animals recognise short and long-chain fatty acids.

*Reviewer #1:*

The revision have improved the quality and impact of the manuscript and opened new questions about the molecular nature of detection of different FA classes and of the circuitry underlying this sensory modality.

There are some typos and run on sentences in the manuscript (i.e line 100, line 352).

An image of the Ir56D GAL4/Gr64f-Lexa; UAS-GCaMP6/LexAop-GAL80 would be useful.

*Reviewer #2:*

The authors set out to address the question as whether the fly's gustatory system can discriminate tastants within the same modality. To some extent, this issue has been addressed already. For example, when flies are offered different sugars at the same concentration, they show different propensities to extend their proboscis (Slone et al. Curr Biol 2007). When flies are given a choice between two carboxylic acids at the same concentration (e.g. acetic acid and lactic acid), they prefer lactic acid (Rimal et al. Cell Reports 2019). Even the responses to octanoic acid (medium chain) and oleic acid (long chain) appear to be different (Kim et al. PLOS Genetics 2018), and this is also documented in an earlier paper from the senior author's lab (Tauber et al. PLOS Genetics 2017). So, the overarching question as to whether flies can discriminate different tastants within the same modality is not completely unexplored. Nevertheless, the authors use a very nice aversive memory assay to show that the flies can discriminate short, medium and long chain fatty acids (FAs), but not different medium chain FAs from each other. They knocked out Ir56d and show that is required for the sensation of medium FAs, consistent with their previous study reporting that Ir56d neurons are required for FA taste. In addition, the Ca^2+^ responses of the Ir56d-expressing neurons were reduced in response in medium chain fatty acids. Unfortunately, the work does not provide a molecular or cellular explanation as to how the flies discern different medium channel FAs. Without such an explanation, the contribution of this work is somewhat incremental over what is already known.

1. Can flies discriminate between different short-chain FAs, and can they discriminate between different long-chain FAs?

2. Lines 156-159: Silencing Ir56d-expressing neurons is not a test of whether Ir56d is required. Nevertheless, the authors have already reported that Ir56d neurons are needed for the response to medium chain FAs so I cannot discern what is new in Figure 3.

3. The authors state that they backcrossed the Ir56dGAL4 mutant to *w1118*. But they do not say for how many generations. 5 generations is typical to eliminate background mutations. At the very least, since they have only one allele, they should confirm the phenotype over a deficiency. The rescue is helpful, but it also changes the genetic background.

4. The authors use GCaMP to examine Ca^2+^ responses to FAs. While GCaMP provides a very good proxy for neuronal activation, the authors should be mindful that rises in Ca^2+^ do not always lead to neuronal activation, and the GCaMP is not the same as measuring action potentials. They should not say that they are measuring neuronal activation. GCaMP allows them to measure neuronal responsiveness, not activation.

*Reviewer #3:*

In this paper, Brown et al. expand earlier observations that fatty acids (FA) at low concentration are detected by neurons expressing Gr64f and IR56d receptor genes. The authors show here that when quinine is associated with FAs of middle range length, flies "learn" not to extend their proboscis to FAs of close length, while they keep extending their proboscis in response to longer or shorter chains. The authors further show that inactivating neurons expressing Gr64f or IR56d prevents proboscis extension to middle range FAs but not longer of shorter FAs. They further create a genetic construction by inserting a Gal4 into the IR56d gene and confirm through calcium imaging experiments, that labellar responses to FAs 6C-8C is mediated by neuron expressing IR56d. From these observations, the authors conclude that flies are able to discriminate between FAs belonging to different categories, ie short-long FAs versus middle length FAs. These data clearly indicate that with the experimental protocol used here, FAs of different length are detected by different populations of gustatory receptor neurons.

However, I am not fully convinced that we have here a clear case of categorical perception. In Masek and Scott 2010 as well as in Kirkhart and Scott 2015, the aversive stimulus was independent of the appetitive stimulus. Here, the situation is more complicated because FAs are mixed with quinine during the training phase.

The conclusions proposed by the authors here rely on the untested assumption that quinine and FAs have no interactions. This might not be the case. Actually, quinine is known to interact with sugar perception (see for ex Meunier et al. 2003), where an exposure to 10 mM quinine (as here) induces an irregular activity in the taste neurons and actually reversibly prevents sugar-sensitive neurons to respond to 50 mM sucrose. Mixing quinine with FAs might thus have a differential effect on gustatory neurons – a repetitive exposure to the mixture might silence the receptors depending on FAs chain length.

Furthermore, hexanoic acid and octanoic acid are known to have a toxic effect on flies. These chemicals are considered as one of the main reasons why noni (the fruit) is toxic to flies except to D sechellia. Earlier observations show that while low doses of these FAs are appetitive, higher doses are deterrent. This means that in the experiments shown here, an additional assumption is that repeated presentations of FAs are not changing the valence of the stimulus. While the toxicity of 6C and 8C FAs is documented, nothing is known about the effects of FAs of a different chain length. Furthermore, to my knowledge, it is not known yet if the deterrent effects of these FAs is due to a disturbance of the responses of neurons responding to sugar or if it activates other populations of gustatory neurons, for example bitter-sensitive.

These two objections could be alleviated if the authors could show data supporting the assumption that gustatory neurons are not changing their responses when in response to consecutive stimulations of FAs and if responses to FAs +quinine mixtures are equivalent irrespective of FAs length.

---

## [Author Response]

[Editors’ note: the authors resubmitted a revised version of the paper for consideration. What follows is the authors’ response to the first round of review.]

The reviewers find fatty acid taste discrimination potentially interesting and agree that the experiments are performed to a high standard. One major concern is whether discrimination is based on intensity rather than quality. A second limitation is that the mechanism of FA detection is not greatly advanced beyond the authors' previous work: the cellular mechanisms for long and short chain FA detection remain unclear. The reviewers agreed that if the major concerns of Reviewer 1 were addressed, this manuscript would provide a broader understanding of fatty acid discrimination.

We are thankful for this thoughtful and critical feedback of our manuscript. We have addressed questions related to intensity through additional behavioral and physiological experiments. In addition, we have expanded on the discussion to highlight the significance of this work over our previous work. In short, we believe the that impact of these studies on discrimination advances our understanding of the general principles of taste coding, extending beyond the neural basis of fatty acid taste. As noted previously, these are some of the most insightful reviews we have received, and we believe that as a consequence, the revised version of this manuscript is substantially improved.

Reviewer #1:This paper investigates fatty acid taste in flies and asks the broad question of whether flies can discriminate different compounds within a single taste modality. The authors' main finding is that flies can discriminate between long, medium, and short chain fatty acids using a previously established aversive memory taste paradigm. When they delve into the cellular and molecular basis of fatty acid detection they find that IR56d neurons respond to all three classes of fatty acids, but are required only for the behavioural responses to medium chain molecules. Similarly, CRISPR/Cas9 deletion of the IR56d receptor reveals that it too is required only for medium-chain fatty acid responses. Thus, different fatty acid classes presumably activate distinct, but partially overlapping subsets of appetitive taste neurons. In general I think the paper is potentially interesting (see comment 1 below) and the data mostly supports the conclusions. However, there is some lack of attention to details that make some of the data hard to interpret.1. The ability of flies to discriminate between different fatty acid classes is presented as the interesting finding, since, as the authors point out, discrimination between compounds within a taste modality is generally not thought to occur. On the surface I agree that this is interesting. However, in the authors' set up of the main question (line 101), they raise an important issue: "Is it possible that flies are capable of differentiating between tastants of the same modality, or is discrimination within a modality exclusively dependent on concentration?" This should be rephrased to replace "concentration" with "intensity" since not all tastants at the same concentration have the same intensity, and from a behavioural perspective it is intensity that matters. Given that, the authors don't do anything to demonstrate that their discrimination task does not depend on intensity, aside from the fact that 1% solutions of all the FA seem to give similar PER. They need to show more explicitly that this task is truly showing identity-based discrimination.

Thank you for these suggestions. We have now replaced the word concentration with intensity. We also include results of new experiments to examine whether discrimination is based on the intensity or the identity of the fatty acid tastants. We find that training flies at a concentration of 1% fatty acid and then testing PER to a different fatty acid at a concentration of 0.1% does not change their ability to discriminate between short-, medium-, and long-chain fatty acids. These data are now shown in Figure 1—figure supplement 2. We now state in the results: “A potential confounding factor in the taste memory assay used to assess discrimination is that flies may discriminate based on perceived intensity, rather than the class identity of fatty acids. To test this possibility, we tested whether flies trained to 1% 6C could discriminate a different fatty acid tested at a different concentration (0.1%). Even under these test conditions, flies remained able to discriminate between short- (5C) and medium-chain (6C) fatty acids as well as between long- (9C) and medium-chain (6C) fatty acids (Figure 1—figure supplement 2A, B). However, conditioning to a medium-chain fatty acid (6C) generalized to another medium chain fatty acid (8C), suggesting that despite a 10x difference in concentration, flies remain unable to distinguish between medium chain fatty acids (Figure 1—figure supplement 2C). These results fortify the notion that flies are able to distinguish between short-, medium-, and long chain fatty acids, but not within fatty acids of the same class” (Line 160).

2. The second broad concern I have is over the nature of short and long chain fatty acid detection. Interpreting the discrimination results would be greatly aided if we knew what other neurons mediate the PER to these molecules. Is it the non-IR56d population of Gr64f neurons? Two experiments would go a long way to addressing this question: (1) silence Gr64f neurons to test whether the broader population is required for short and long chain FA PER and (2) do calcium imaging of Gr64f neurons and see whether non-IR56d projections (which according to the authors' previous work are spatially segregated in a more dorsal area of the SEZ) respond to short and long chain FA.

Thank you for this suggestion. We have now examined PER in flies with silenced *GR64f* neurons. In agreement with previous data, we find that *GR64f*-GAL4>*UAS-TNT* flies have reduced PER to medium-chain (6C-8C) fatty acids. We also find that short- (4C-5C) and long-chain (9C-10C) fatty acid response remains intact. Therefore, while *IR56d* neurons, and therefore a subset of *GR64f* neurons, are responsive to short- and long-chain fatty acids, these neurons are dispensable for the behavioral response. These findings suggest that neurons outside the canonical sweet-sensing neurons that detect both sugars and fatty acids are responsible for PER to short- and long-chain fatty acids. These data are now shown in Figure 3D-E. Further, imaging of neural activity in non-*Gr64f*-expressing *IR56d* projections revealed that these neurons are responsive to short-, medium-, and long-chain fatty acids. These data are now shown in Figure 6.

Reviewer #2:In the present paper Brown et al., study the ability of *Drosophila melanogaster* to discriminate between Fatty Acids (FAs) of different lengths. Using a combination of behavioral experiments, molecular biology and in vivo calcium imaging, the authors show that a subset of Ir56d expressing neurons are able to differentiate FAs. However, the Ir56d receptor is only necessary for the detection of medium-length FAs but not short- or long-. The paper explores in detail the role of the Ir56d receptor as FA detector, a role previously described by the authors in a previous paper Tauber et al. 2017.I consider that the experiments are properly done, and so the statistical analysis, however gain in knowledge is very limited. So far, the authors can prove that flies can discriminate FAs of different lengths, being Ir56d the receptor detecting medium-length FAs, a result that expands the knowledge gained in Tauber et al. 2017. In figure 3, the authors show that silencing Ir56d neurons using tetanus toxin expression, reduces dramatically PER to medium-length fatty acids, but not to short or long, pointing to a different set of neurons involved in their detection.

We have now included data silencing all *GR64f*-expressing neurons (Figure 3D-E). We find that silencing these neurons reduces PER to medium-chain fatty acids, but not to short- or long-chain fatty acids, bolstering support for the conclusion that these classes of fatty acids are also detected by other populations of neurons.

However, the in vivo calcium imaging experiments show that Ir56d neurons also respond to short- and long- FAs. In this regard, I disagree with the statement at the abstract: Characterization of hexanoic acid-sensitive Ionotropic receptor 56d (Ir56d) neurons reveals broad responsive to short-, medium-, and long- chain fatty acids, suggesting selectivity is unlikely to occur through activation of distinct sensory neuron populations. In fact, I consider that selectivity would come from the activation of different subsets of gustatory neurons. It seems that Ir56d neurons could be a subset of the neurons that generally respond to FAs, providing the specificity for medium-length FAs. Other neurons, in addition to the Ir56d ones might be responding to short- and long- FAs in an Ir56d independent manner.

Thank you for these comments. Indeed, we agree with this notion. In the initial statement we meant to convey that selectivity is unlikely to occur through different subsets of *IR56d* neurons rather than other subsets of gustatory neurons. We have made this correction. We now state: “While *IR56d* neurons are broadly activated by short-, medium-, and long-chain fatty acids, genetic deletion of *IR56d* selectively disrupts response to medium-chain fatty acids. Further, *Ir56d+Gr64f+* neurons are necessary for PER to medium-chain fatty acids, but both *IR56d* and *GR64f* neurons are dispensable for PER to short- and long-chain fatty acids, indicating the involvement of one or more other classes of neurons.” (line 32). In addition, we note in the revised manuscript that the anterior and posterior projecting populations of *IR56d* neurons have different responses to short-, medium-, and long-chain FAs, providing an additional mechanism of selectivity. Overall, we understand that our results do not fully address the question of selectivity. However, our observations of differences in taste representation of fatty acid groups provide clear paths to investigate the underlying mechanisms.

I consider the authors should explore in deep how short- and long- FAs are actually detected, whether it depends on other Ionotropic Receptors (probably Ir25a and Ir76b might be involved (Ahn et al. 2017)) and which subset of gustatory neurons are actually responding to these compounds, considering they do not require Ir56d nor Ir56d neurons.

We have now included data measuring taste response to short-, medium-, and long-chain fatty acids in both *IR76b* and *IR25a* mutant flies (Figure 3 – Supplemental Figure 1). We find that taste responses to all fatty acids tested are reduced in both mutants, suggesting that these receptors, in addition to *IR56d*, are required for response to medium-chain fatty acids, but that additional, yet unidentified, receptors are required for response to both short- and long-chain fatty acids. We now state in the results: “Since *IR56d* is required for taste response to medium-chain, but not short- or long-chain fatty acids, it is possible that other IRs mediate this response. As a first step in identifying which additional receptor(s) may be involved, we measured PER in both *IR76b* and *IR25a* mutants to short-, medium, and long-chain fatty acids, as these broadly expressed receptors have been previously found to mediate taste response to medium-chain fatty acids (Ahn et al., 2017). In agreement with these findings, PER to medium-chain fatty acids were significantly reduced for both the *IR76b* and *IR25a* mutants, while PER to sucrose was normal (Figure 3—figure supplement 1). Additionally, we found that PER to both short- and long-chain fatty acids were also significantly reduced in both mutants, suggesting that both *IR76b* and *IR25a* are required for taste response to all three classes tested (Figure 3—figure supplement 1).” (line 219). As noted above, we also include results showing that PER to short- and long-chain fatty acids remain unaffected in *GR64f*-GAL4>*UAS-TNT* flies, and imaging data to show that GRNs expressing *IR56d* but not *GR64f* respond to short-, medium- and long-chain fatty acids.

Reviewer #3:[…]Overall comments and questions:1. Are the differences in taste discrimination between male and female flies?

We now include data for PER to short-, medium-, and long-chain fatty acids in male flies. In comparison to females, we find that PER is generally lower in males. We also measured taste discrimination in male flies and found that, similar to females, males are also able to differentiate between short-, medium-, and long-chain fatty acids. We now state in the results: “We next sought to determine whether the discrimination observed in female flies is also found in males. PER analysis with a panel of short-, medium-, and long-chain fatty acids revealed that males respond to all fatty acids tested, though the overall responses were lower than those observed in females (Figure 1—figure supplement 3A). To assess whether male flies are able to discriminate between different classes of fatty acids, we trained flies to a medium-chain fatty acid (6C) and then measured discrimination between 4C (short), 8C (medium), and 9C (long) fatty acids. Similar to results obtained in female flies, males were able to discriminate between 6C and 4C as well as between 6C and 9C, but not between 6C and 8C (Figure 1—figure supplement 3B-D). Therefore, male flies are also able to discriminate different classes of fatty acids, but are not able to distinguish fatty acids within the same class” (line 172).

2. Individual data points should be shown whenever possible for all figures (except PER because that would make it impossible to interpret).

We have added individual data points to the appropriate figures.

3. Can the authors discuss how discriminating between different fatty acids types may be adaptive? Are they found in different food sources, some of which are "good" and some "bad"? Is there evidence from other organisms about this type of molecular discrimination in fatty acid taste?

Thank you, this is an excellent suggestion and an interesting topic for us to explore. We now include a paragraph in the discussion addressing this matter, stating: “Fatty acids are natural by product of yeast fermentation (Diwan and Gupta, 2018; Nyanga, et al., 2013; Oliveira et al., 2011), and their abundance in fruit declines after ripening (Duan et al., 2013). Further, fatty acids have antifungal activity, which scales with chain length (i.e the greater the chain length, the greater the antifungal efficiency; (Pohl et al., 2011)). (Pohl et al. 2011). Thus, the ability to discriminate between different classes of fatty acids is likely be important in determining the stage of fruit ripeness, degree of fermentation, and the general palatability of a potential food source/oviposition site.” (line 336).

[Editors’ note: what follows is the authors’ response to the second round of review.]

The observation that short- and long-chain PER responses are independent of Gr64f neurons suggests that the discrimination between medium vs. short- or long-chain FA may actually be no more surprising (in retrospect) than the discrimination between medium-chain FA and sugars (reported previously). In both cases, the "other" compound can elicit PER through GRNs that are not activated by MC-FA. Therefore, one could speculate that in both cases, the flies learn that IR56d+ GRN activation is "bad" but still perform PER because LC-FA, SC-FA, or sucrose all elicit PER without IR56+ GRNs (even though they all also activate those GRNs).

Indeed, we agree with these comments, however, we believe they highlight the novelty of this system. As reviewers stated, previous work on sugars concluded that flies are unable to discriminate within an individual modality (Masek and Scott, 2010). The differences described suggest neural circuits involved in discrimination are dissimilar from those typically studied as being required for PER. Therefore, we hope this manuscript (and Tauber et al., 2017) position flies as a model to study taste discrimination in addition to innate feeding response. Towards this end, we now state in the discussion: “Our findings highlight the complexity of taste discrimination, which extends beyond simple PER as a readout for taste. For example, all types of fatty acids tested increase GR64f neural responsiveness, however, only GR64f neurons are required for PER to medium-chain fatty acids, thereby raising the possibility that short-, medium-, and long-chain fatty acid taste discrimination occurs through different neural channels. These findings stress the need to define the fatty acid receptors and neural circuits that govern responses to short- and long-chain fatty acid taste.” (line 375).

Reviewer #1:The revision have improved the quality and impact of the manuscript and opened new questions about the molecular nature of detection of different FA classes and of the circuitry underlying this sensory modality.There are some typos and run on sentences in the manuscript (i.e line 100, line 352).

We have proofread for typos and run on sentences and have modified the manuscript accordingly.

Reviewer #2:The authors set out to address the question as whether the fly's gustatory system can discriminate tastants within the same modality. To some extent, this issue has been addressed already. For example, when flies are offered different sugars at the same concentration, they show different propensities to extend their proboscis (Slone et al. Curr Biol 2007). When flies are given a choice between two carboxylic acids at the same concentration (e.g. acetic acid and lactic acid), they prefer lactic acid (Rimal et al. Cell Reports 2019). Even the responses to octanoic acid (medium chain) and oleic acid (long chain) appear to be different (Kim et al. PLOS Genetics 2018), and this is also documented in an earlier paper from the senior author's lab (Tauber et al. PLOS Genetics 2017). So, the overarching question as to whether flies can discriminate different tastants within the same modality is not completely unexplored. Nevertheless, the authors use a very nice aversive memory assay to show that the flies can discriminate short, medium and long chain fatty acids (FAs), but not different medium chain FAs from each other. They knocked out Ir56d and show that is required for the sensation of medium FAs, consistent with their previous study reporting that Ir56d neurons are required for FA taste. In addition, the Ca^2+^ responses of the Ir56d-expressing neurons were reduced in response in medium chain fatty acids. Unfortunately, the work does not provide a molecular or cellular explanation as to how the flies discern different medium channel FAs. Without such an explanation, the contribution of this work is somewhat incremental over what is already known.

We hope our comment above, that “our findings highlight the complexity of taste discrimination, which extends beyond simple PER as a readout for taste” sufficiently emphasizes the difference between taste discrimination and innate preference assays. We also now state in the discussion: “Our aversive taste memory assay confirmed previous findings that flies can discriminate between sugars and fatty acids (Tauber et al., 2017), and led to the surprising observation that flies can distinguish between different classes of fatty acids, even though the baseline responsiveness to short- medium-, and long-chain fatty acids were similar in innate preference assays.” (line 359).

1. Can flies discriminate between different short-chain FAs, and can they discriminate between different long-chain FAs?2. Lines 156-159: Silencing Ir56d-expressing neurons is not a test of whether Ir56d is required. Nevertheless, the authors have already reported that Ir56d neurons are needed for the response to medium chain FAs so I cannot discern what is new in Figure 3.3. The authors state that they backcrossed the Ir56dGAL4 mutant to w1118. But they do not say for how many generations. 5 generations is typical to eliminate background mutations. At the very least, since they have only one allele, they should confirm the phenotype over a deficiency. The rescue is helpful, but it also changes the genetic background.

The IR56d^GAL4^ mutant line was backcrossed for 10 generations. We now include in the methods: “All lines were backcrossed to the *w1118* fly strain for 10 generations.” (line 463).

4. The authors use GCaMP to examine Ca^2+^ responses to FAs. While GCaMP provides a very good proxy for neuronal activation, the authors should be mindful that rises in Ca^2+^ do not always lead to neuronal activation, and the GCaMP is not the same as measuring action potentials. They should not say that they are measuring neuronal activation. GCaMP allows them to measure neuronal responsiveness, not activation.

We understand this comment. We did not intend to imply that GCaMP was reflective of action potentials. We have modified the term ‘neuronal activation’ to ‘neuronal responsiveness’ or ‘taste-evoked changes in Ca^2+^’ where appropriate.

Reviewer #3:In this paper, Brown et al. expand earlier observations that fatty acids (FA) at low concentration are detected by neurons expressing Gr64f and IR56d receptor genes. The authors show here that when quinine is associated with FAs of middle range length, flies "learn" not to extend their proboscis to FAs of close length, while they keep extending their proboscis in response to longer or shorter chains. The authors further show that inactivating neurons expressing Gr64f or IR56d prevents proboscis extension to middle range FAs but not longer of shorter FAs. They further create a genetic construction by inserting a Gal4 into the IR56d gene and confirm through calcium imaging experiments, that labellar responses to FAs 6C-8C is mediated by neuron expressing IR56d. From these observations, the authors conclude that flies are able to discriminate between FAs belonging to different categories, ie short-long FAs versus middle length FAs. These data clearly indicate that with the experimental protocol used here, FAs of different length are detected by different populations of gustatory receptor neurons.However, I am not fully convinced that we have here a clear case of categorical perception. In Masek and Scott 2010 as well as in Kirkhart and Scott 2015, the aversive stimulus was independent of the appetitive stimulus. Here, the situation is more complicated because FAs are mixed with quinine during the training phase.

The FAs are not mixed with quinine during the training phase. They are each applied independently to the proboscis. We have clarified this in the manuscript. In the results we now state: “We used an aversive taste memory assay in which an appetitive tastant applied to the proboscis is paired with application of bitter quinine immediately afterwards, resulting in an associative memory that inhibits responses to the appetitive tastant (Masek et al., 2015).” (line 115). We have also clarified this in Figure 1A. We now state: “Next, flies were trained by pairing this fatty acid with quinine presentation immediately following tastant application (Training).” (line 780).

The conclusions proposed by the authors here rely on the untested assumption that quinine and FAs have no interactions. This might not be the case. Actually, quinine is known to interact with sugar perception (see for ex Meunier et al. 2003), where an exposure to 10 mM quinine (as here) induces an irregular activity in the taste neurons and actually reversibly prevents sugar-sensitive neurons to respond to 50 mM sucrose. Mixing quinine with FAs might thus have a differential effect on gustatory neurons – a repetitive exposure to the mixture might silence the receptors depending on FAs chain length.Furthermore, hexanoic acid and octanoic acid are known to have a toxic effect on flies. These chemicals are considered as one of the main reasons why noni (the fruit) is toxic to flies except to D sechellia. Earlier observations show that while low doses of these FAs are appetitive, higher doses are deterrent. This means that in the experiments shown here, an additional assumption is that repeated presentations of FAs are not changing the valence of the stimulus. While the toxicity of 6C and 8C FAs is documented, nothing is known about the effects of FAs of a different chain length. Furthermore, to my knowledge, it is not known yet if the deterrent effects of these FAs is due to a disturbance of the responses of neurons responding to sugar or if it activates other populations of gustatory neurons, for example bitter-sensitive.

In the experiments performed here, toxicity is highly unlikely because flies were not allowed to ingest the FAs with the assay system we utilized, in which a wick saturated with tastant contacts the proboscis to stimulate proboscis extension. We have clarified this in the methods. We now state: “A wick made of Kimwipe (#06-666; Fisher Scientific) was placed partially inside a capillary tube (#1B120F-4; World Precision Instruments; Sarasota, FL) and then saturated with tastant, thereby enabling flies to taste, but not ingest tastant.” (line 501). In response to repeated presentations of FAs not changing the valence of the stimulus, we find that behaviorally, PER to consecutive applications of FA show no significant change in responsiveness over time. We now state in the results: “Further, it is possible that repeated presentation of fatty acid may alter the valence of the tastant over time. To address this, we asked whether there were any differences in PER between the pretest, training, and test applications among the naïve groups in each test of taste discrimination. We found that PER to consecutive applications of fatty acid show no significant change in responsiveness over time, although in some cases PER to fatty acid trends downward (Figure 1—figure supplement 1).” (line 154).